# A Taxonomy of Challenges to Curating Fair Datasets

**Dora Zhao**[*]
Stanford University

**Morgan Klaus Scheuerman**[*]
Sony AI

**Pooja Chitre**[†]
Arizona State University

**Jerone T. A. Andrews**[†]
Sony AI

**Georgia Panagiotidou**
King's College London

**Shawn Walker**[‡]
Arizona State University

**Kathleen H. Pine**[‡]
Arizona State University

**Alice Xiang**[‡]
Sony AI

## Abstract

Despite extensive efforts to create *fairer* machine learning (ML) datasets, there remains a limited understanding of the practical aspects of dataset curation. Drawing from interviews with 30 ML dataset curators, we present a comprehensive taxonomy of the challenges and trade-offs encountered throughout the dataset curation lifecycle. Our findings underscore overarching issues within the broader fairness landscape that impact data curation. We conclude with recommendations aimed at fostering systemic changes to better facilitate fair dataset curation practices.

## 1 Introduction

Persistent concerns from academia, government, industry, and the public sphere center on the disparate impact and unfairness in machine learning (ML) [22, 28, 32, 69–71, 74, 77, 94, 143, 159]. Data is often viewed as a primary culprit, perpetuating biases and compromising fairness [36, 52, 88, 164]. In response, substantial attention has been directed towards *fair* dataset collection practices [43, 46, 62, 65, 114, 121, 135, 161, 164]. However, there remains a significant gap in understanding both the practices and practicalities of fair dataset curation.

To address this gap, we shift from theoretical, guideline-focused scholarship [3, 41, 42, 51, 63, 78, 80, 103, 117] to empirical inquiry, exploring the grounded practices of fair dataset curation. Following a well-established tradition in human-computer interaction (HCI) [67, 75, 97, 107, 125, 149], we conducted interviews with 30 dataset curators from both academia and industry who have experience curating fair vision, language, or multi-modal datasets. Through these interviews, we uncover practical challenges and trade-offs to ensuring fairness in dataset curation. Our use of qualitative methodology allowed us to surface nuanced challenges and trade-offs that regularly appear throughout the curation process and gain insights into considerations that may otherwise remain undisclosed.

We first provide three dimensions of fairness—*composition*, *process*, and *release*—that participants considered during curation. Fairness is not only a property of the final artifact—the dataset—but also a constant consideration curators must account for throughout the curation process. Through our empirical findings, we identify various challenges that obstruct different fairness goals. Building on Hutchinson et al. [78]'s conception of the dataset lifecycle, we contribute a taxonomy of challenges dataset curators encounter, addressing both dataset lifecycle-specific challenges (Section 3) and those within the broader landscape of fairness in ML (Section 4). By conducting in-depth interviews

---

[*]Joint first author

[†]Joint second author

[‡]Joint last author

38th Conference on Neural Information Processing Systems (NeurIPS 2024) Track on Datasets and Benchmarks.

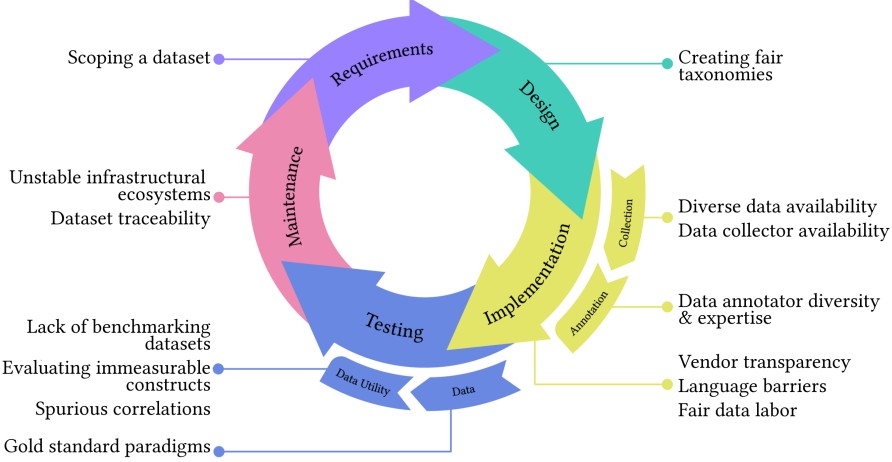

Figure 1: A circular process diagram showing how each challenge we identified maps to each phase and subphase of the dataset lifecycle.

with those engaged in fair dataset work on the ground, we provide empirical support for prior work [3, 42, 55, 78–80, 91, 93, 111], which has focused on identifying implicit challenges in the fairness literature (see Appendix B for additional background). We conclude with recommendations aimed at fostering systemic changes to better facilitate fair dataset curation practices (Section 5).

Our work aligns with existing recommendations for fair dataset curation [3, 12, 42, 51, 78, 80, 98, 102, 103, 111, 117, 131] and aims to deepen stakeholders' understanding of the specific challenges involved. By illuminating these issues, we hope to expedite more effective solutions and promote further investigation into the complexities of fairness in dataset curation.

## 2 Method

To understand the challenges of collecting fair datasets, we conducted 30 semi-structured interviews with ML dataset curators, each lasting between 45–60 minutes, between November 2023 and March 2024. Participants were asked to define fairness in ML datasets, describe their process for collecting fair datasets, highlight challenges encountered, and discuss any trade-offs made. Refer to Appendix A for more details, including Institutional Review Board approval.

**Participants.** To qualify, participants must have previously curated at least one fair ML dataset. Given the extensive discourse surrounding language and vision dataset practices [12, 16, 111], we prioritized participants in these domains. To accommodate diverse perspectives, we refrained from prescribing a specific definition of "fair." Initial recruitment was conducted through purposive sampling [142], targeting authors of public datasets, followed by outreach via social media and relevant mailing lists, with snowball sampling [142] used to expand participation.

To protect anonymity, participants are referred to as "PX", where "P" denotes "Participant" and "X" represents their identification number (e.g., P8).

**Thematic Analysis.** To analyze the interviews, we adopted an inductive approach [20]. We began with an initial set of codes derived from our literature review on challenges in fair data collection. Four authors independently coded the same interview to identify additional themes, refined the codebook through discussion, and repeated the process with a second interview. The remaining interviews were then equally distributed among the research team for thematic analysis.

## 3 Challenges During the Dataset Lifecycle

We present challenges participants encountered across the dataset lifecycle, taxonomizing them into requirements, design, implementation, evaluation, and maintenance phases (see Figure 1 and Table 3).

Recognizing the multi-faceted nature of *fairness*, we did not impose a specific definition during our interviews. Instead, we empowered participants to articulate their own definitions. Based on these definitions, we identified three dimensions of fairness: *composition*, which is achieved through diverse representations; *process*, which includes equitable compensation for data subjects and workers as well as recognition for curation efforts; and *release*, which emphasizes the importance of transparent and openly accessible data. The challenges we surface span all three dimensions of fairness.

## 3.1 Requirements

The *requirements phase* involves establishing a dataset's purpose (e.g., intended tasks such as image tagging) and defining the fairness criteria to be operationalized within the dataset (e.g., group fairness). Challenges in this phase most often manifested in the composition and process dimensions.

**Scoping a dataset.** Participants sought to balance fairness with utility (P8, P23, P26, P30). On the one hand, careful curation can lead to more nuanced insights compared to general-purpose datasets. As P26 explained, they would ideally "*design smaller datasets for smaller models for specific applications, nothing that is deployed on a [South Asian] scale, because that definitely won't work properly because of the [region's] geographical diversity.*" Moreover, datasets containing billions of entries, such as LAION [133, 134], make oversight difficult and, as a result, may include "*unfair*" data (P18) [15]. Nonetheless, participants also had to consider utility. P13 noted ML is "*in this age of scale,*" making them "*a bit skeptical as [to] whether people are going to openly use fair datasets for training unless they're very large.*" P21 highlighted a similar tension between "*technical reasons why you need large open datasets*" and "*ethical reasons on why that shouldn't be the case.*" Fairness trade-offs pushed some (P12, P13) towards focusing on smaller evaluation datasets.

**Determining fairness definitions.** Nearly all participants stressed the *contextual* nature of fairness. Key factors shaping their definitions included domain (e.g., healthcare), task (e.g., sentiment analysis), and cultural context. For example, P2 highlighted the importance of cultural specificity, stating, "*you see a lot of work that talks about fairness in gender or in race. But for a [South Asian] country, race does not manifest like it manifests for America.*" Participants also made trade-offs due to the multitude of fairness definitions available [104] (Section 4.5). P19 noted that "*there's more than two dozen different fairness definitions ... used in the literature.*" This diversity necessitated sacrifices in other dimensions, as emphasized by P18, who illustrated this with the "'*no free lunch theorem*'", stating, "*You can't have complete diversity with respect to, say, races,...geographies,...times of the day, and other domains. Everything is not possible. Once you clamp on one, the other one goes away.*"

## 3.2 Design

In the *design phase*, curators determine how to operationalize dataset requirements, including defining the dataset's taxonomy. For example, curators specify attributes for measuring fairness (e.g., skin tone) and the categories within those attributes [66, 68, 114, 145]. This phase also involves decisions on data collection and annotation methodologies (e.g., web scraping, hiring vendors). Challenges in this phase typically arose in the composition and process dimensions.

**Creating fair taxonomies.** Participants struggled to find a fair taxonomy under the inherent unfairness of categorization. For example, P18 devised a geographic taxonomy featuring categories for the U.S. and Asia, acknowledging that the regions "*are not homogeneous, they're very heterogenous.*" P2 also noted a theoretically ideal taxonomy is as granular as possible, but practical constraints, such as data availability (Section 3.2) and time (Section 4.3), necessitated using coarser categories. Finally, the challenge of creating a fair taxonomy was compounded by the inadequacies of existing domain taxonomies. For example, P1 and P5 pointed out that the common binary operationalization of gender in medical data erases many gender identities. Nonetheless, participants felt compelled to utilize inadequate taxonomies due to practical constraints, even if it contradicted their personal beliefs. Participants were forced to align their notions of fairness with disciplinary norms (Section 4.2).

**Data availability in taxonomy design.** Similar to when designing taxonomies, participants had to balance their ideal data collection methods with practical constraints. For example, P3's dataset only included Spanish and Arabic even though they "*wanted to look at other languages, but ... didn't have training data.*" Participants questioned prevailing data collection paradigms, such as web scraping [3], which were seen as unethical when performed indiscriminately. For legal compliance, P25 manually

collected data for two years: *"I was downloading, like clicking and clicking, because they didn't allow me to do web scraping or didn't have an API."*

## 3.3 Implementation

The *implementation phase* marks the execution of plans from the design phase, where curators collect, annotate, and package the data into a dataset. This phase broadly encompasses two subphases: data collection and data annotation. Challenges in this phase span all three dimensions of fairness.

### 3.3.1 Data Collection

*Data collection* involves gathering relevant data to fulfill dataset requirements. Challenges during this subphase prevented participants from attaining fairness goals relevant to dataset diversity.

**Diverse data availability.** Similar to concerns raised regarding dataset taxonomies (Section 3.2), participants raised concerns about data availability for creating a fair dataset. For example, P28 described how sexist stereotypes permeate web data, such as *"women [being] associated with nurse more often than men."* Additionally, P18 encountered difficulties sourcing web data from *"Middle Eastern"* and *"African countries"* but found *"lots and lots and lots of images from India, Japan and [the] U.S., which are like the three most dominant geographies in uploading pictures."* Participants also lamented the inaccessibility of specialized or proprietary data, such as medical records or data from private companies, which could significantly improve the creation of fair datasets. P4 stated that *"because people don't own large e-commerce platforms or social media platforms, or whatever, we just kind of have to deal with things that we can gather from existing systems."*

Interestingly, synthetic data, sometimes presented as a potential solution to biased data [6, 120, 141], was met with skepticism as it could perpetuate stereotypes or inadequately represent underrepresented groups [156]. As P19 pointed out, *"You might address some of the missing data points [with synthetic data] but at the end of the day it's still the same underlying data distribution, right?"*

**Data collector availability.** Many participants associated fairness with geographically diverse data. For example, P22 expressed how they would *"proactively sample more data from underrepresented regions."* Yet, actualizing this objective proved challenging, as P12 highlighted the difficulty in *"get[ting] hold of people ... from very, very small regions."* Infrastructure hurdles, such as limited internet and mobile phone access, further complicated the process [3, 80]. Equipping data collectors with necessary equipment is costly (Section 4.3) and logistically challenging, as *"you might have to give people smartphones to start and you'd also need more labor on the ground ... who are working in these different regions to come together and do this"* (P12).

### 3.3.2 Data Annotation

*Data annotation* involves labeling data with attributes specified during design. Participants faced challenges recruiting annotators who had requisite expertise or came from diverse backgrounds. Upholding fair labor practices (Section 3.3.3) during annotation also presented challenges.

**Data annotator diversity and expertise.** The interpretation and application of annotation categories can vary based on an annotator's perspective [2, 7, 25, 81]. P22 described finding annotators for labelling building styles across different geographies: *"You give this same image to a local labeler who is in that culture, who is an expert in, you know, their architecture ... then you get a much better label."* Yet, participants had difficulty hiring annotators that met their desired aims. While P2 highlighted the value of diverse annotator backgrounds or beliefs to ensure annotations reflected a wide range of experiences, accessing diverse annotators was challenging, *"because some of the attributes of [annotators'] personal lives might even be illegal to ask about in a particular country."* Participants also confronted challenges in recruiting annotators with specialized expertise. For example, despite offering *"$75 or $100 per hour,"* P1 faced difficulties finding and incentivizing medical experts to annotate radiology data. Annotators who lack diversity or expertise in data concepts may lead to issues with data quality, including inaccuracies [76], biases [44, 48, 127], and overly homogeneous annotations [48, 115]. Notably, P13 highlighted that crowdsourced annotators regularly embed gender biases into datasets such that *"researchers [need] to make sure that annotators represent everyone because [if] not, you're just gonna have a skewed pool of annotations as well."*

### 3.3.3 Implementation Processes

Participants expressed challenges not only with dataset content but also with the *implementation* of data collection and annotation. We provide three main considerations discussed by participants.

**Vendor transparency.** Collaborating with data vendors introduced transparency challenges, hindering fairness efforts. First, as prior research documented [128], vendors may prohibit access to data worker identities, such as demographic details or location (as described by P2 in Section 3.3.2). Thus, it is impossible to evaluate potential biases or expertise linked to identity characteristics, such as how an annotator's cultural identity may influence their engagement with data concepts. Second, participants had little oversight into worker compensation or encountered communication restrictions imposed by vendors. As P12 said, *"I think [pay] was fair in terms of [being] calibrated across different countries ... but we weren't able to get exact numbers, because that was confidential."* P6 described how vendor platform design inhibited direct collection of feedback from data workers, impeding efforts to improve fairness in dataset creation and labor conditions (e.g., [101, 102]) (Section 4.5).

**Language barriers.** Curating fair datasets often involves collecting geographically diverse data, which may require data workers proficient in languages different from those of curators. Language barriers can hinder effective communication, necessitating fairness concepts established in the design phase (Section 3.2) to be accurately translated into the workers' native languages. Improper translations can result in misinterpreted labels or instructions and may even lead to contract breaches, particularly concerning subject consent. Addressing language barriers often involves resorting to translation services, which may be constrained by cost (Section 4.3) or introduce its own fairness concerns. Further, participants had to ensure translations accurately reflected their intentions, but as P3 noted, *"We relied on our translators to come up with those sorts of decisions in terms of Spanish."*

**Fair data labor.** Several participants (P6, P11, P12, P14, P16, P24, P28) expressed concern about engaging in fair labor practices when working with data workers, but systemic organizational (Section 4.3) and regulatory (Section 4.4) issues made achieving these standards difficult.

## 3.4 Evaluation

The *evaluation phase* involves assessing data quality and testing dataset utility. Challenges in this phase can result in homogeneous annotations, benchmarking difficulties, and spurious correlations, most often affecting the composition and release of a dataset.

### 3.4.1 Assessing Data Quality

*Assessing data quality* entails validating and refining the data and its annotations to ensure clarity and consistency with project requirements. (Re)alignment of data and annotations with the guidelines from the design phase is often referred to as *quality assurance*.

**Gold standard paradigms.** Participants often sought to capture a diversity of perspectives across annotators. Thus, prevailing practices for validation and cleaning, such as majority voting and annotator agreement metrics, may be unsuitable. As P24 emphasized, majority voting can "*squash or stifle diverse opinions when it comes to subjective tasks.*" When disagreement is integral to the objective, annotator agreement metrics become inappropriate, making it difficult to "validate" annotation quality. Gold standard paradigms are intrinsically tied to disciplinary challenges (Section 4.2); if submitting a publication involving dataset creation, reviewers might still call for annotator agreement metrics and believe the quality of the data is poor if agreement is low.

Similarly, common practices used to clean or filter data can perpetuate dominant cultural beliefs. Data that might appear noisy or incorrect can hold significance for certain communities. P14 explained how quality filters resulted in "*get[ting] rid of vernacular that's not perfect English but is maybe like African-American vernacular or like Hispanic-American vernacular, and that also introduces bias and lowers the diversity of the dataset.*" This echoes prior work [9] which found that standard data filters might disproportionately exclude content from already marginalized groups.

### 3.4.2 Evaluating Data Utility

To ensure dataset utility, curators must evaluate its effectiveness, often through *requirements testing* to confirm its suitability for the intended purpose. Participants aimed to align the dataset with fairness definitions and mitigate any potential biases present in the data.

**Lack of benchmarking datasets.** Curators often seek to benchmark their datasets to showcase their utility. However, since many participants aimed to create unprecedented fair datasets to address existing gaps, this norm posed a challenge as comparable datasets were non-existent. Reflecting on the struggles with a novel geodiverse dataset, P12 explained, "*We couldn't measure it unless we had a dataset that actually was fair. Since we don't have a dataset that is fair..., you are arguing in circles.*" Furthermore, even if comparable datasets exist, they may harbor fairness issues of their own.

**Evaluating immeasurable constructs.** Evaluating whether a dataset aligns with fairness definitions presupposes that fairness is a construct amenable to measurement. While some participants offered quantifiable indicators of fairness, such as demographic diversity, others argued that fairness defies quantification. P14 criticized measurement-oriented perspectives, stating, "*They also assume that fairness can be measured, can be evaluated, and can be improved. And I think that all of this is a more positivist mindset.*" Even with a definition in mind, testing may feel incomplete. As P28 said, "*Even when you provide a way to measure fairness, you're probably overlooking something.*"

**Spurious correlations.** Several participants (P6, P23, P28) aimed to avoid introducing spurious correlations that affected the fairness of the dataset's composition [24, 53, 82]. While these correlations may not be "*connected with any demographic or social variable*" (P23), they can still influence downstream models and result in biased decisions. However, as recent research [100] has revealed, spurious correlations with demographic attributes are ubiquitous. Thus, enumerating and removing all possible correlations is virtually impossible.

### 3.5 Maintenance

In the *maintenance phase*, curators must consider both how their dataset is released and strategies for ensuring its ongoing utility over time. Challenges at this stage often linked back to participant concerns around fairness in dataset release (i.e., ensuring the data is transparent and openly accessible).

**Unstable infrastructural ecosystems.** Digital data is intrinsically impermanent. Some participants (P1, P8, P30) emphasized the risk of data instances disappearing due to broken links or shifts in platform popularity or ownership, as observed with platforms like Twitter. Therefore, curators must then not only monitor for missing data but also find suitable replacements that match the original dataset's distribution. This can be particularly burdensome when the data was expensive (Section 4.3) or difficult to collect (Section 3.3.1). As data goes missing, datasets can become unbalanced and thus "unfair," demonstrating how fairness issues with data release are linked to concerns about composition.

**Dataset traceability mechanisms.** The challenge of dataset stewardship is exacerbated by inadequate traceability mechanisms [112, 132]. Participants underscored their inability to track users and usage patterns of their datasets. One commonly used proxy is citations in academic papers, but it was hard to "*distinguish citations that use the data versus citations that use the broader idea of the paper*" (P2). This is concerning, especially if fair datasets are repurposed in unintended ways. While prior works [3, 112, 132] have suggested data usage policies to mitigate such risks, enforcing them becomes impractical when curators are unaware of actual data users.

## 4 Challenges Overarching the Broader Landscape of Fairness

The dataset curation process is influenced by the environments in which curators operate, meaning their decisions are not made in isolation. Many challenges span all phases of the lifecycle, shaping the broader landscape of dataset fairness. We identified five levels within this landscape, where challenges may emerge from one or more levels, affecting dataset curation at every phase of the dataset lifecycle (see Figure 2 and Table 4).

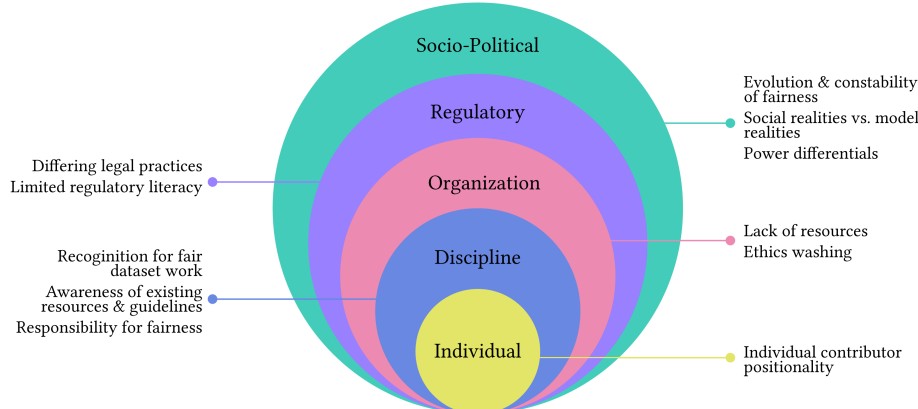

Figure 2: A social ecological [21] representation of challenges in each layer in the overarching landscape of fairness. A social ecological model shows how each layer is nested but interconnected.

## 4.1 Individual Level

The *individual level* of the dataset curation landscape refers to the contributors of fair datasets, such as data curators, data subjects, and data workers.

**Individual contributor positionality.** Decisions made by contributors were inevitably influenced by their own unique perspectives [126, 129]. As P24 said, *"There's this stuff we swim in that we don't really realize is even there."* Despite recognizing this influence, assessing its tangible impact on the dataset remained elusive. Addressing and diversifying contributor positionality is further complicated by other challenges within the dataset curation landscape, such as cost and power differentials. Positionality was evident in instances where participants felt they had to make trade-offs during processes like designing taxonomies that may erase others' experiences. P27 encouraged reflecting on personal values: *"Is this [research] actually in line with your life philosophy? Was it in line with your gender, with your sexuality... If it's not, would you still want to be doing this?"*

## 4.2 Discipline Level

The *discipline level* of the dataset curation landscape centers on the norms and practices governing specific academic disciplines, particularly ML [14, 45, 118, 131].

**Recognition for fair dataset work.** Despite the growing demand for data in ML, according to participants, fair dataset curation efforts were not seen as significant contributions to the field. P11 described a *"lack of general disciplinary value of datasets as contributions."* While some major conferences like NeurIPS [148] have introduced dataset tracks, few venues prioritize dataset-focused work. This lack of appreciation discourages efforts to ensure dataset stability and longevity [131].

**Incentive mechanisms.** Incentives in ML do not align well with the costs of fair dataset curation. According to P11, there's *"just [a] total lack of resources and time to actually deeply engage with labeling and sourcing those labels and getting people who are representative of those labels to be the data workers."* Participants echoed well-documented observations that model work is valued over data work [124, 125, 131], with P21 stating that *"data is kind of a second-class citizen in ML research."* Consequently, P25 felt *"people are [not] seriously talking about fairness ... people are still just get[ting] whatever [data] they get to do their research, or publish, or whatever."*

**Awareness of existing resources and guidelines.** Participants had limited awareness of existing guidance for fair dataset curation. This lack of awareness may be attributed to some of these resources (e.g., [42, 78, 80, 117]) being disseminated outside of traditional ML venues (e.g., NeurIPS, *CL, ICML, CVPR). As P29 admitted, *"I don't remember any explicit guidelines that I've stumbled through for fair dataset collection. Honestly!"* Promoting interdisciplinary awareness of fairness efforts among those primarily involved in ML is challenging due to highly disciplinary norms that prioritize novelty in ML methods over discussions on fair dataset curation.

**Responsibility for fairness.** The burden of responsibility for fairness weighs most heavily on individuals aware of fairness concerns in ML. Participants echoed findings from prior research [14] that document how fairness is not a top priority for many ML researchers. For example, P25 said that "*[in] the team I work with... I never heard them talking about [how] the dataset has to be fair.*" In P25's experience, the norm was to cursorily engage with fairness issues without substantive changes to research practices. Given the lower prioritization of fairness in ML, the onus falls on individual researchers who "*have a strong sense of justice and fairness*" (P24) or are part of fairness-oriented communities to elevate these concerns. However, this commitment often lacks external recognition and may hinder resource allocation and research progress. Participants recognized that collecting fair data is more challenging and resource-intensive compared to conventional methods: "*If you want to build a fair dataset, maybe the most efficient way to do that is to scrape the web, but getting really diverse data in an ethical way is really hard and really expensive*" (P11).

### 4.3 Organization Level

The *organization level* refers to the organizations where individuals conduct fair dataset curation work, which could vary in size or nature, such as academic or industry settings.

**Lack of resources.** Insufficient resources were a significant challenge across all phases of the dataset lifecycle. As P1 declared: ''*Money?! (laughs) If you have money, you can have a very high quality of data.*" Fair data collection methods are costly, especially concerning data quality and annotation, which often require hiring experts. Convincing funders or stakeholders of the value of investing in fair datasets proved difficult, as noted by P24: "*It's hard to convince somebody to spend thousands and thousands to collect [a] dataset of recordings.*" Moreover, participants aimed to compensate data subjects and workers fairly, "*not just the minimum wages that many times academia gives*" (P29). Longterm maintenance costs added to the financial burden, with difficulties in securing ongoing funding. P1 stated no academic or industry organization "*[wants] to spend another millions of money every year ... to maintain those products.*"

**Ethics washing.** Participants disapproved of organizations that superficially promote fair ML but fail to meaningfully integrate fairness into their practices [151]. According to P16, the "*language of fairness is simply external lip service [that] ultimately boils down to looking at the maximization of other imperatives, such as economic ones.*" Resource constraints exacerbate this issue, leading organizations to prioritize efficiency and cost-effectiveness over fairness. As P22 noted, "*A lot of big companies do responsible AI shenanigans ... for marketing ... And then a new shiny thing comes down the road, and then they join that instead.*" When fairness is valued primarily for its marketing appeal rather than its impact on product development, it is not prioritized for monetary or labor investment.

### 4.4 Regulatory Level

The *regulatory level* concerns laws and policies governing dataset curation and use. Participants expressed anxieties about violating regulations they were not necessarily equipped to fully understand.

**Differing legal practices.** Contextual laws and regulations posed a challenge for participants. P2 described how "*laws in America or laws in Europe ... might not be directly applicable to a [South Asian] country that has a very different societal situation.*" Contextually contingent laws and policies further complicated efforts to obtain data from diverse, underrepresented populations (Section 3.3.1).

**Legal risk.** Throughout the dataset lifecycle, participants faced the looming risk of unintentionally violating laws and regulations, potentially leading to breaches of privacy, labor, or data ownership laws. Instances of inadvertent violations are not uncommon, as highlighted by participants' experiences with web scraping practices. For example, P21 was aware that "*people discovered links to child pornography*" in a widely used benchmark dataset [16, 144]. In another instance, P5 described working on a clinical dataset only to learn that releasing it was "*not possible because it's not consistent with the privacy laws in France.*" To mitigate these risks, some participants adopted highly cautious practices, such as exclusively collecting royalty-free or Creative Commons images, and storing only image URLs to avoid any copyright violations. However, these strategies can result in dataset instability, as observed by P8, who faced issues with broken URLs.

**Limited regulatory literacy.** Insufficient understanding about navigating the law intensified concerns about legal risk. P8 described it as "*a big learning curve to understand what we were allowed to store*

*and what we weren't.*" As a result, P8 consulted an intellectual property lawyer. However, depending on the other constraints dataset curators are under, such as discipline (Section 4.2) or organization (Section 4.3) level constraints, hiring legal counsel may be untenable.

## 4.5 Socio-Political Level

The *socio-political level* covers the shifting social and political contexts around fairness in which curators operate. These challenges can be conceptualized as thorny, fluid, and arguably insoluble.

**Evolution and contestability of fairness.** According to P3, fairness will *"always be up for debate,"* making it *"sort of impossible for there to be like a gold standard."* Fairness is subjectively perceived, influenced by individual contexts, experiences, and beliefs [129]. This subjectivity fuels ongoing scholarly debates [42, 89, 137]; it also fueled diverse perspectives among participants. As P30 pointed out, *"There are people from the audience who say that we have a good definition [of fairness], and there are some people who say that we have a terrible definition. And there's no way to make everyone happy."* The absence of a universally accepted definition complicated participants' efforts to operationalize fairness in dataset curation. Further, existing guidelines may not suit every notion of fairness, leading to divergent curation methodologies. As P14 highlighted, *"It's kind of like a philosophical question ... while the quantitative method says that fairness can be achieved, contrast it to qualitative that we are just trying to understand the experience here."* Beyond disagreements about what fairness means (or should mean), participants also noted that current definitions are not stable. As P16 put it, fairness *"should be a notion that is able to evolve within society, and certain forms of injustice that were not considered injustice[s] in the past now are ... there might be other evolution towards the future that we currently do not incorporate in our definition of fairness, and we need to account for that."* This perpetual evolution presents challenges for dataset curators. They must decide whether to regularly update datasets or retract them as definitions evolve. However, both approaches have limitations in addressing the continued use of previously released datasets [93, 112].

**Social realities versus model realities.** P8 described how the real world is different than "*what's experimentally valid and testable.*" Due to the complexity of the real world, certain groups inevitably remain underrepresented, misrepresented, or overlooked entirely despite best efforts. For example, P12 mentioned that while they wanted to collect images from underrepresented countries, data collector availability constrained their options (Section 3.3.1). Participants also questioned whether balanced representation was even the best approach. As P1 pointed out, *"The problem is when you actually apply such a model to the real world, the real world is imbalanced, right?"* This echoes the classic trade-off between fairness and accuracy in algorithmic fairness work [31]. Curators must wrestle not only with the impossible task of how to best account for every human experience in a dataset, but also whether or not they should be.

**Power differentials.** Power imbalances contribute to fairness issues during the curation process that are not visible in the dataset's composition. Participants noted how more elite institutions and companies dominate efforts to create fair datasets, largely owing to their access to resources (Section 4.3). Similar to findings from prior work [85], P21 described how most public datasets are not used, with the majority of *"the datasets that get used in ML research [being] created by a very, very small elite cadre of ... academic institutions that have close affiliations with top industry researchers."* Similarly, P16 felt it was problematic that the *"most important tools"* remain in the hands of a few companies, *"yet they are given the freedom to define what is fair, and their definition is used, and then the safeguards that do exist might not always align or ensure protection.".* Thinking on a geopolitical scale, P2 noted that *"the field of algorithmic fairness has been dominated by the Western perspective."* This imbalanced representation exacerbates other challenges previously outlined, including those at the implementation, disciplinary, and organizational levels.

Power differentials also permeate relationships between dataset curators and other stakeholders, including data subjects and workers [152]. For example, P6 described how curators have complete oversight over worker compensation: *"So many platforms don't actually ensure that you're fairly compensating workers. And it's really up to the individual researchers which is a crazy system that sets absolutely the wrong incentives."* P10 compared the impulse to collect data cost-effectively, at the expense of data subjects, as *"a particular kind of colonial impulse, like, this is just up for grabs."* Similarly, curator decisions have profound implications downstream. P22 described the difficulty of *"fighting"* clients who do not prioritize model performance on heavily under-resourced populations, given they are not central to business incentives: *"It's like, '99% of my customer[s] will be fine, why*

*do I need to care about that last 1%?"* Overall, dataset curation was seen as *"a very unfair process, no matter how you do it ... unless you're going to literally tackle society"* (P8).

## 5    Recommendations for Enabling Fair Dataset Curation

Finally, we highlight recommendations across the three dimensions of fairness for facilitating fair dataset curation. We focus on top-down efforts, reflecting the need for systemic changes rather than relying solely on individual contributions. See Appendix D for additional recommendations.

**Composition.** To better enable fair dataset *composition*, we encourage interventions for more flexible and robust data practices. For example, at the design phase (Section 3.2), flexible taxonomies can facilitate different operationalizations rather than forcing curators to use only one taxonomy (e.g., protected attributes can include self-reported and third-party labels). At the discipline level (Section 4.2), we advocate for more communication across academic communities. Papers published outside traditional ML venues (e.g., CHI, FAccT, CSCW) have provided guidance on data curation, such as annotation practices [30, 42, 83, 155] or considerations on taxonomies [7, 64, 84, 160].

**Process.** A change in the fair dataset curation *process* requires not only norm-setting within fairness communities, but also legal and policy interventions. For example, at the implementation phase (Section 3.3.3), participants were concerned about labor rights for data workers. As a discipline, we should have norms about compensating workers, at least at the local minimum wage, for their labor and support efforts to introduce policies that offer codified protection for data workers. Furthermore, at the regulatory level (Section 4.4), rather than expecting curators to develop legal expertise, we advocate for the creation of accessible resources on legal practices regarding dataset collection.

**Release.** We encourage interventions that allow for fairness post-*release*. For example, at the maintenance phase (Section 3.5), efforts to build tools and policies to enable better dataset traceability could alleviate concerns with dataset misuse. Additionally, at the organization level (Section 4.3), funding entities should invest in maintenance, rather than solely focusing on modeling research. Monetarily valuing long-term maintenance plans as research contributions may help shift perspectives about revision, maintenance, and use policies at the discipline level (Section 4.2).

## 6    Discussion and Conclusion

Our qualitative data reflects the experiences of our participants, and while we identify shared themes, these challenges may not be universally applicable or entirely representative. Despite efforts to recruit diverse dataset curators, our sample is skewed toward curators from North America and Europe, reflecting the Western-centric nature of ML and fairness research [85, 138]. Given the challenges raised around creating culturally contextualized datasets and navigating power dynamics across regions, future work should aim to include more geographically diverse voices, especially from the Global South, for deeper, more nuanced insights.

Despite these limitations, our study offers an important foundation for addressing the practical challenges in fair dataset curation. Through interviews with dataset curators engaged in fair dataset work, we developed a taxonomy of challenges across the dataset lifecycle and the broader fairness landscape. Participants navigated complex trade-offs between ideal fairness goals and practical constraints such as data availability, resources, and time. While we acknowledge the limitations of our methodology, taxonomizing these challenges is a crucial first step in developing long-lasting solutions to support fair dataset curation.

Addressing these challenges will require effort not only from individual dataset curators but also systemic changes at organizational, disciplinary, and regulatory levels. Beyond providing dataset curators with grounded evidence to support their efforts in building fair datasets, our taxonomy offers stakeholders a pathway to address each challenge individually and opens avenues for further, more targeted investigations into the many challenges of curating fair datasets.

## Acknowledgments and Disclosure of Funding

This work was funded by Sony Research.

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

# A Methods

## A.1 Participant Recruitment

We interviewed 30 ML dataset collectors, refining our protocol through two pilot tests before recruitment. Out of 204 individuals contacted, we received 95 no responses, 51 declines, and 28 who did not meet the criteria. Recruitment concluded with 30 participants, reaching thematic saturation. As shown in Tables 1 and 2, participants represent diverse backgrounds and experiences, with a predominant presence from academia. Compensation consisted of a $75 Amazon gift card, or the equivalent in the participant's local currency.

| Type | Count |
|------|-------|
| Role | Graduate student (13), Post-Doctorate Researcher (6), Faculty (4), Researcher [Industry] (3), Researcher [Institute-based] (2), Other (2) |
| Setting | University (23), Industry (4), Academic Research Institute (2), Think Tank (1) |
| Modality | Language (16), Vision (9), Multi-modal (5), Tabular (3) |
| Location | Northern America (19), Southern Europe (3), Western Europe (3), Northern Europe (2), Latin American & the Caribbean (1), Western Africa (1), Southern Asia (1) |

Table 1: Summary statistics of participant demographics. The locations are coded at the region level according to the United Nations geoscheme. Since some participants had experience collecting datasets in more than one modality, the counts in this row exceed 30.

## A.2 Participant Anonymity

At the beginning of the interview, participants were asked to provide their informed consent. They were given the option to opt-out of the interview and also told they have the right to withdraw from the study at any time. Participants were also asked for permission to record the study over Zoom. For data protection, each interview was transcribed from the Zoom recording and identifying details—including but not limited to names, institutions, and dataset names—were redacted from the interview transcript before the coding process. To preserve participant anonymity, participant recruiting and interviews were conducted only by members of the research team from Arizona State University. Only the redacted interviews were shared with other members of the research team for analysis.

## A.3 Thematic Analysis

We also provide additional details on our thematic analysis protocol. After establishing an initial codebook of themes, the research team (N=4) independently coded one of the interviews. We then reconvened and synchronously discussed how we coded the interviews and analyzed where we differed when applying codes. After this initial coding round, we again independently coded a second interview and repeated the same process of discussing any disagreements amongst the team before creating a finalized codebook. Only after reaching agreement on the definitions and applications of codes did we split up the remaining interviews amongst the team members.

To identify themes from the code, we had each member of the research team first generate themes, with supporting quotations, they observed in the interviews. Then, the research team met synchronously over four sessions to discuss and distill these observations into the higher-level themes discussed in the paper.

Finally, to ensure thorough consideration, we drew on a diverse range of expertise by following contemporary interdisciplinary practices [119, 123]. Our team consists of researchers, practitioners, and lawyers with backgrounds in HCI, ML, CV, algorithmic fairness, health sciences and policy, data visualization, and social and behavioral science. With varied ethnic, cultural, and gender backgrounds,

we bring together extensive experience in dataset design, model training, and the development of ethical guidelines.

| | Participants | |
|---|---|---|
| *Participant ID* | *Organization Type* | *Dataset Focus* |
| P1 | Academia | Language |
| P2 | Academia | Language |
| P3 | Academia | Language |
| P4 | Academia | Other |
| P5 | Academia | Language |
| P6 | Industry | Language |
| P7 | Industry | Language |
| P8 | Academia | Multi-modal |
| P9 | Academia | Language |
| P10 | Academia | Language |
| P11 | Academia | Vision |
| P12 | Academia | Vision |
| P13 | Industry | Vision |
| P14 | University | Vision, Language, Other |
| P15 | University | Vision |
| P16 | Academia | Other |
| P17 | Academia | Multi-modal |
| P18 | Academia | Vision |
| P19 | Academia | Other |
| P20 | Academia | Vision |
| P21 | Academia | Language |
| P22 | Industry | Language, Vision |
| P23 | Academia | Language, Multi-modal |
| P24 | Academia | Language |
| P25 | Academia | Language |
| P26 | Academia | Vision |
| P27 | Academia | Language |
| P28 | Academia | Multi-modal |
| P29 | Academia | Language |
| P30 | Academia | Multi-modal |

Table 2: A of participants we interviewed for this study. Organization type refers to whether participants were in academia or industry. Dataset focus refers to the type of data participants collected for their dataset. "Vision" refers to visual data such as images and/or videos. "Language" refers to natural language data, such as textual data and/or spoken language data. "Multi-modal" refers to datasets which included both vision and language data. "Other" refers to datasets that fall outside of this schema, such as tabular datasets.

## A.4  Interview Protocol

We provide the protocol used to guide the semi-structured interview process conducted with participants. The interview questions were designed based on considerations around fair dataset curation that had been raised in the existing literature. Depending on the answers that the participants provided, the interviewers asked relevant follow-up questions. The questions are as follows:

- Please briefly describe your current role and responsibilities. What way(s) does your current role interface with dataset collection for machine learning?

- What is the role of machine learning in your organization?

- What types of data do you collect to train and/or evaluate ML algorithms? What are the sources of this data?

- Do you have any processes or are you currently developing any processes to ensure the fairness of data collected and used to train and/or evaluate ML algorithms?

- How does your organization define "fairness" of datasets? Do you have a formal, codified definition of fairness?

- How did your organization decide on the definition for fairness? Which factors influence this?

- How do you ensure collection of fair datasets to train and/or evaluate ML algorithms? Or fairness when repurposing collected datasets?

- Can you walk me through the process of making data collection and or data sets fair, as you do and experience it?

- Which best practices did you employ to ensure the collection or making of fair datasets?

- Which factors, in your experience, influence the making/collection of fair datasets?

- What challenges did you experience during the process of making/collecting datasets?

- How did you handle those challenges?
  - What were some workarounds/ solutions?
  - If you cannot recall any challenges, what about the process made it relatively smooth / why do you think there were not challenges?
  - Were any parts easier or more difficult than expected?

- Thinking back to the process of making or collecting datasets, I'd like you to tell me a story about a time when you experienced any trade-off related to fairness of the dataset during that process — meaning, you had to sacrifice something to increase the fairness of the dataset, or you sacrificed fairness to achieve something else.

- What challenges has your organization had in maintaining fairness in your datasets?

- Since collecting fair datasets, have you released any of these datasets?

- Thinking beyond your specific domain, what items should be included in more general guidelines for the creation and maintenance of fair datasets to train and/or evaluate ML algorithms? Are there any gaps in our current practices?

- Do you have any comments or other points to make? Is there anything we did not cover in the interview which you would like to talk about?

- Do you have any suggestions/advice about who we should talk to next?

# B  Background

Concerns over the disparate impacts or unjust outcomes associated with machine learning (ML) continue to persist [22, 28, 74, 94, 143]. One of the central concerns underscoring the pursuit of fair ML remains the datasets used to develop ML systems [12, 13, 22, 92, 111]. Yet obtaining fair and ethically-sourced datasets remains a challenge. Data is often perceived as the scourge of ML models and a source of for downstream biases [36, 52, 88, 164]. Here, we provide background on

prior scholarship documenting the current issues with dataset curation, as well as work focused on improving those practices.

**Issues with existing dataset curation practices.** Poor training data can lead to representational harms [22, 50, 73], such as stereotyping [19, 23, 49, 136], spurious correlations [100, 154, 165], and poor performance or total erasure of certain populations [22, 158, 164]. Poor evaluation data means harmful model outcomes may be overlooked or missed, especially as they cascade into various (often unintended) domains [125]. Beyond data's impact on models directly, ML datasets are increasingly scrutinized for violating the ethical values of privacy and consent [3, 35, 105, 111, 113], reinforcing disputable social constructs [10, 17, 64, 84, 130], including highly offensive content [12, 15, 164], and exploiting vulnerable populations for both data and annotations [59, 140, 152].

Practices for collecting large-scale data, such as web scraping, have consistently failed to meet many legal standards at the local and national level, violating copyright laws [61], biometric laws [72, 163], and even including child exploitation content [16, 144]. The difficulty of authoring and maintaining a comprehensively "fair" dataset is exacerbated by differential definitions of fairness and how to measure it (or whether it can be measured at all) [4, 99, 146]. Current approaches to dataset documentation also obscure the inherently collaborative work that dataset authors must engage in and negotiate [101, 102].

**Improving dataset curation practices.** Given the vast and varied issues with ML datasets, there has been a extensive line of work focused on improving dataset collection practices. These efforts have evolved substantially beyond *ante hoc* calls for more transparent and robust documentation of existing datasets [8, 29, 42, 51, 98, 109, 117], such as datasheets, which often result in a *ante hoc* approach, thus failing to capture decisions and trade-offs which might have occurred prior to and during data collection. Thus, scholars are attempting to provide frameworks at different levels of granularity of considering the responsibility of dataset authors leading to frameworks or design guidelines for both *pre hoc* and *per hoc* dataset curation [3, 54, 58, 107, 110, 131, 157].

For example, at a higher level, Scheuerman et al. [131] proposed a value-centric framework that centers values like positional expertise and contextually-relevant annotations. Andrews et al. [3] released a comprehensive set of considerations for responsibly curating human-centric computer vision datasets, covering topics like consent, human diversity, and subject revocation. Recent work from Orr and Crawford [107] distilled seven recommendations from interviews with 18 dataset curators. Their work highlights high-level themes such as advocating for more dataset auditing, ensuring participant privacy, and encouraging more documentation. Scholars are also increasingly providing highly contextual and specific guidance for collecting data on certain subgroups and vulnerable populations, such as children [72, 153, 157], who are increasingly ending up in large web-scraped datasets [16, 144].

The scholarship focused on providing considerations and guidance for ethical dataset curation has been invaluable. However, how authors actually approach curating fair datasets is still opaque—especially given documented gaps between guidance and practice. Prior work has uncovered numerous barriers to incorporating fairness into practice [38, 67, 75, 96, 97, 151], including misalignments between available toolkits and product needs [38], organizational trade-offs that make auditing methods less effective [39, 106], and difficulty negotiating expectations across roles [40, 96]. Yet literature on the challenges to creating fair datasets currently lacks a holistic framing of fairness that involves not only the composition of datasets, but the practices of producing and maintaining them. Identifying the challenges currently facing dataset curators focused on creating fair datasets is crucial to enabling fairer dataset curation in both industry and academic settings.

## C  Additional Figures and Tables

To illustrate each of the challenges we identified, in Table 3, we provide an example quotation from our interviews. We similarly map out the overarching landscape of fairness challenges using a social ecological model [21] and provide detailed examples (see Figure 2 and Table 4).

**Taxonomy of Challenges to Creating Fair Datasets**

| Phase | | Challenge(s) | Definition | Example |
|---|---|---|---|---|
| Requirements | | Scoping a dataset | Determining the size and scope of the dataset and its taxonomy while remaining true to fairness goals | *"If the face images are on a billion scale, there's no way we can identify each of this person in real world to reach out to them ask if they are okay with it."* (P18) |
| | | Determining fairness definitions | Deciding which definition of fairness to adopt and which not to | *"Other work does not explicitly define fairness and that assumption can change the way you look at something. And so, being explicit about your intentions and about your working definitions and the inclusions and the exclusions of the scope of work, ... is getting attention now more."* (P2) |
| Design | | Creating fair taxonomies | Establishing a system of classification that aligns with fairness definitions, despite classifications being inherently imperfect | *"There are power relationships within what the model represents and what it represents is hegemony and it represents rigidity and classification, and it does not represent all of that queerness."* (P27) |
| | | Data availability in taxonomy design | Designing taxonomy with knowledge of data (un)availability in mind | *"The basic challenge is actually the availability of the data ... So, you need to set up boundaries in your research. I discuss the certain limitation on this issue in [our] paper. We don't have information on gender because the healthcare system does not adopt non-binary gender attributes."* (P1) |
| Implementation | | Vendor transparency | Working with data vendors can hinder fairness efforts | *"Communication is difficult on platforms like [ANONYMIZED VENDOR PLATFORM]. It's a little bit easier on other platforms, but that has definitely been a blocker. It is like easy communication just doesn't exist."* (P6) |
| | | Language barriers | Navigating language barriers between curators and data workers and/or data | *"So, we relied on our translators to come up with those sorts of decisions in terms in terms of Spanish. We did. Some of us didn't know Spanish, not natively and fluently."* (P3) |
| | | Fair data labor | Ensuring data workers are treated and compensated fairly while navigating resource and regulatory constraints | *"Especially, because so many platforms don't actually ensure that you're fairly compensating workers. And it's really up to the individual researchers which is a crazy system that sets absolutely the wrong incentives."* (P6) |
| | *Data Collection* | Diverse data availability | Difficulties collecting sufficiently diverse or representative data during the data collection process | *"Because when I look at images of stoves on the Internet..... The reason those images are up on the Internet is because they satisfy something other than someone's going to use to train a machine learning model. So, people put them up because they think it's something new or exciting, or something different in some way. So, a lot of stoves that are there for ImageNet are basically like product images from stores that they sell. So, the stoves always look very clean and new and things like that."* (P12) |
| | | Data collector availability | Identifying data collectors who can collect underrepresented data | *"Because it very much depends on where we can get hold of people, right? And by that, I mean where it is up and has its workforce. And also how much money [does it cost] because it becomes more expensive as you're trying to get a lot of people from very, very small regions."* (P12) |
| | *Data Annotation* | Data annotator diversity and expertise | Identifying data annotators with situated expertise | *"The challenges in actually getting diverse annotators. Especially for like, let's say, there have been recent studies, or there has been one paper that talks about how the views of a person or a kind of how the views of the person affect the annotations that they do for hate speech. So, like people with certain social, certain political viewpoints, might annotate something as hate while others might not. And so how do you get diversity in your annotators? Because some of the attributes of their personal lives might even be illegal to ask about in a particular country ... And once you have it, how do you contextualize their annotations to their lived experiences?"* (P2) |

| Phase | | Challenge(s) | Definition | Example |
|---|---|---|---|---|
| Evaluation | Data Quality | Gold standard paradigms | Models for assessing dataset quality can promote "unfairness" | "There is no ground truth to that question. It can vary from a person's lived experiences to the next. So, it is inherently a subjective question. So, we did not want to squash those annotations down to a majority quote." (P2) |
| | Data Utility | Lack of benchmarking datasets | Comparable benchmark datasets for which to evaluate new fair datasets are not available | "There's a lot of pressure to do well benchmark data sets. And so, there's a risk of them being used overused because you need to show that you did well on the data set that everyone recognizes, even if it might not be the most appropriate." (P21) |
| | | Evaluating immeasurable constructs | Proving dataset quality when fairness constructs are not quantifiable | "I think quantitative methods almost always assume that fairness can be achieved in some way and they also often assume that there is already a robust definition of fairness that we've conceptualized and that we can use to test our systems. They also assume that fairness can be measured, can be evaluated and can be improved. And I think that all of this is a more positivist mindset." (P14) |
| | | Spurious correlations | Accounting for and controlling spurious correlations | "So, then, what happens is there is a geography bias which is being incurred in this data sets implicitly, which is not really explicit. I'm gonna train the models on this. The models just exaggerate the bias and when this model is deployed on, say, android phones, or software or laptops, or anything, the consumers are worldwide, right?" (P18) |
| Maintenance | | Unstable infrastructural ecosystems | Data in datasets may go missing or become deprecated, resulting in fairness issues | "I think maintenance is more going to be a matter of making sure that when links become deprecated, we maintain the same principles of trying to find a diverse range of images to replace it." (P8) |
| | | Dataset traceability mechanisms | Inability to track dataset usage or prevent misuses | " there have been cases where a researcher reached out to me and said, 'Hey, I tried this with your data set. I'm getting like these confusing results. Can we talk?' And then I find out they're using it in a way that wasn't intended." (P6) |

Table 3: A table describing each of the challenges throughout the phases of the dataset lifecycle.

| Phase | | Challenge(s) | Definition | Example |
|---|---|---|---|---|
| *Overarching* | *Individual* | Individual contributor positionality | Every contributor to a dataset has their own positionality, including biases | *"Even the idea of the perspectivism ... Most obviously in my work is the research questions, and then the way it informs the direction of research, and even possibly down to the way we qualify how good a data set and how interesting a dataset is!"* (P24) |
| | *Discipline* | Recognition for fair dataset work | Datasets are undervalued in machine learning | *"The right way is also rewarding people for doing it the right way right like the idea that you should be able to publish a data set and that be a valuable contribution, because in machine learning, it's an extremely valuable contribution. And yet it's not something that is valued."* (P21) |
| | | Awareness of existing resources and guidelines | Curators are unaware of existing resources for fair datasets or how to apply them | *"If I recall like, I don't like remember any explicit guidelines that I've stumbled through for fair data set collection. Honestly!"* (P29) |
| | | Responsibility for fairness | Those with an awareness about fairness issues feel a responsibility to do fairness work, while those who are not aware are excused from fairness work | *"In general, I will say the motivation is having fairness because you have this responsibility of understanding and improving transparency and improving general oversight on what we deploy."* (P28) |
| | *Organization* | Lack of resources | Fair dataset work is not given resources in the form of time, money, personnel, tools, etc. | *"Research is driven by building bigger and bigger models and that is increasingly, punitively expensive. From a resource standpoint, from a money standpoint, from an environmental standpoint. And data has, in general, been undervalued in machine learning.* (P21) |
| | | Ethics washing | Fairness is treated as a marketing tactic rather than necessary | *"One of the big reasons a lot of big companies do responsible AI shenanigans is for marketing ... then a new shiny thing comes down the road and then they join that instead."* (P22) |
| | *Regulatory* | Differing legal practices | Laws, regulations, and policies governing fairness differ by context | *"Laws in America or laws in Europe ... might not be directly applicable to a country like [in South Asia] that has very different societal situation."* (P2) |
| | | Limited regulatory literacy | Dataset curators are not equipped to understand the regulatory landscape | *"It was a big learning curve to understand what we were allowed to store and what we weren't in terms of the legal sense. So, it was a challenge to us personally, because we didn't have experience. So, we consulted with an IP lawyer to get insight into that, but really just making sure that what we were presenting and storing was legal."* (P8) |
| | *Socio-Political* | Evolution and contestability of fairness | Perspectives and policies on fairness evolve over time, constantly evolving the landscape of what a fair dataset is | *"The question [of] whether fairness should be defined through a singular definition within a specific instrument is tricky because ... It should be a notion that is able to evolve within society, and certain forms of injustice that were not considered injustice in in the past now are. If we looked at the position of members of the LGBTQIA+ community, it was criminalized. Racism was also accepted. Now we clearly say it's not so. There might be other evolution towards the future that we currently do not incorporate in our definition of fairness, and we need to account for that."* (P16) |
| | | Social realities versus model realities | The "real" world is inherently complex and multifaceted, but machine learning datasets (and downstream models) require more simplistic approaches | *"Benchmark[s] which are made for fairness ... still have a very structured, kind of neutral way of portraying things like race or gender that don't actually engage with the socio-historical meaning of that.* (P11) |
| | | Power differentials | Different institutions (e.g., industry vs. academia; elite universities vs. R3s), actors (e.g., data curators vs data workers), and regions (e.g., the West vs the Rest) have different power to shape fairness concepts and practices | *"I mean you hear of data coming from these marginalized regions but then this is centralization process with one institution getting credit for it and the reputations of other countries not sharing that credits and some not reaping benefits of it. So, there's especially in countries that are poorer, there's then less incentive for them to actually contribute to datasets."* (P8) |

Table 4: A table describing each of the challenges overarching the broader landscape of fairness

# D    Detailed Recommendations for Enabling Fair Dataset Curation

Recommendations are aimed at diverse stakeholders influencing fair dataset curation, including—but not limited to—individual contributors, academic institutions and venues, industrial organizations, policymakers, and the affected public. Unlike in the main body of the text, where we describe the challenges with the dataset lifecycle first and the challenges with the overarching landscape of fairness second, here, we present considerations with the overarching landscape foremost. We also begin with the highest level of the dataset landscape, the *socio-political level*, rather than the lowest, the *individual level*. Our goal is to underscore how top-down changes can have broader impacts downstream on individual data curators and the dataset lifecycle. We advocate for more systemic changes rather than placing the onus of fairness onto individuals. The following recommendations in Appendices D.1 and D.2 are examples. We imagine there are many more interventions which would be effective in improving fair dataset curation.

## D.1    Recommendations Overarching the Broader Landscape of Fairness

### D.1.1    Socio-Political Level

**Evolution and contestability of fairness.**  As conceptualizations of fairness inevitably change, curators should aim to keep datasets up-to-date. For example, we recommend that curators revise and amend datasets to comply with new conceptualizations of fairness. For example, Yang et al. [162] obfuscated faces in ImageNet after release as an effort to mitigate concerns about data subject privacy. Furthermore, when datasets cannot be aligned with new standards, norms, laws, or policies surrounding fairness, they ought to be deprecated and no longer used. Curators can refer to Luccioni et al. [93]'s framework for retracting and deprecating datasets to better understand this process.

We also recommend that data curators clearly document the decisions that were made about contextually and temporally relevant definitions of fairness. Thus, even if the original curator cannot afford to update the dataset, others can continue to maintain its documentation pointing toward new research showing the issues with past fairness operationalizations.

**Social realities versus model realities.**  We recommend dataset curators engage with affected communities to understand the needs and potential impacts datasets and downstream models have on the lives of real people. This includes situating data curation decisions in the experiences and perspectives of affected communities. For example, Kuo et al. [86] introduced WikiBench, a system for creating community-driven evaluation dataset on Wikipedia. Using WikiBench, community members can work together to select, label, and discuss instances for an evaluation dataset. Adopting a more participatory and bottom-up approach allows dataset curators to ensure that they are capturing the concepts most relevant to impacted communities.

**Power differentials.** First, we recommend incentivizing dataset curation with fairness perspectives outside of the West and Global North [26, 138]. For program committees or conference chairs, potential actions can include having special tracks for these datasets or offering travel scholarship for researchers to the conference. We advocate for approaches that empower researchers from the Global South to create their own datasets.

Another power differential participants discussed was between researchers and data subjects or annotators. To address this, we urge curators to center the agency and consent of data subjects as well as the expertise of data workers. Rather than treating data workers as "ghost workers" [59], curators should ensure that data workers are meaningfully involved throughout the data curation process and thought of as contributors rather than solely as a labor source.

### D.1.2    Regulatory Level

To help minimize legal risk, our first recommendation is for the the discipline to develop ethical review processes to assess for potential legal implications of dataset collection. Venues, such as NeurIPS, have instituted impact statements and paper checklists for submitted works [11]. We recommend that this reviews extends to include legal risks. We advocate for this discipline-wide approach as it can defray potential concerns regarding resource mismatches when it comes to consulting legal counsel. By developing a standardized procedure for legal compliance across datasets, it also reduces the burden on individual curators who may have limited regulatory literacy.

Nonetheless, we still recommend that individual curators pay particular care when collecting data containing people or about people. One alternative here, such as that taken by Asano et al. [5] and Ramaswamy et al. [121], is to ensure there are no people in the dataset. Of course, there is still a need for human-centric datasets. In this case, we urge curators to recognize that using royalty-free or Creative Commons licenses does not absolve the data of potential ethical or legal issues regarding privacy or consent [3]. Instead, curators ought to obtain informed consent from data subjects following well-established protocols from human subjects research [47].

### D.1.3  Organization Level

**Ethics washing.** Participants were disillusioned by organizations that treated fairness as "lip service" and engaging in the practices of ethics-washing. Echoing Wang et al. [151], we recommend institutional efforts to keep organizations liable for the ethical AI promises that they make. In addition to relying on individual contributions from researchers and journalists, having watchdog organizations monitor for ethics-washing. This recommendation draws from existing practices of monitoring companies for "greenwashing", or manipulative promises from companies that they are engaging in environmentally friendly actions [34].

### D.1.4  Discipline Level

**Lack of recognition and incentives.** Since 2021, there have been efforts to introduce more dataset-focused tracks, such as the Datasets and Benchmarks track at NeurIPS [1] or the Journal of Data-Centric Machine Learning Research (DMLR). We recommend building on this trend and encouraging more dataset-focused tracks, including some that have specific sub-areas dedicated to fairness-oriented datasets. This can help address the lack of recognition and incentives for fair dataset work.

**Responsibility for fairness.** Fairness-oriented changes ought to be widely adopted amongst ML dataset creators, not only those who may be more "fairness' or "justice" oriented. To encourage this shift, we recommend adopting more educational training on these subjects. Universities can include fairness and ethics into computer science courses. An example of this are the Embedded EthiCS programs at universities which encourage students to think critically about the technology they are learning about in their computer science courses [60]. Beyond university courses, AI ethics review processes can also mandate certifications that researchers must complete prior to getting approval similar to the trainings that researchers must complete before receiving IRB approval.

### D.1.5  Individual Level

**Individual contributor positionality.** Contributor biases are inevitable. Our recommendations here focus not on removing all individual biases but rather on encouraging curators to get multiple perspectives and reflect on what biases they may be bringing prior to data collection. One recommendation is to institute a "pre-registration" system similar to what social scientists have in place [108]. Pre-registration requires social scientists to publicly state their hypotheses, methods, data collection process, and analysis plans prior to beginning their experiment. Filling out a pre-registration prior to data collection could encourage curators to think through design biases and justify the choices they have made in a transparent and standardized manner.

## D.2  Recommendations During the Dataset Lifecycle

### D.2.1  Requirements

**Determining fairness definitions.** Participants considered fairness to be highly contextual. To ensure that the definitions of fairness match those of impacted communities, we recommend that curators solicit and incorporate community feedback into the design and evaluation of fairness criteria [18]. This can help ensure that the dataset reflects the needs and values of diverse populations. As an example for how this can be done, curators can look to works such as Shen et al. [139] which aimed to involve community members in deliberative processes for defining AI systems. Similar participatory processes can be adapted for determining fairness definitions in datasets.

### D.2.2 Design

**Creating fair taxonomies.** When designing a label taxonomy, we encourage curators to evaluate trade-offs associated with adopting coarser categories, such as loss of granularity versus feasibility and practicality. Data curators should report both their ideal data collection scenario and the actual approach taken. This information is useful not only from a transparency perspective but also for other researchers who may face similar issues in the future.

In addition, these taxonomies should be designed with scalability in mind. Curators should make provisions to ensure the taxonomy is flexible enough to incorporate new data if collected. For example, the OpenImages dataset [87] has had several new versions and additions since its initial release, including Schumann et al. [135]'s new demographic annotations which are aimed to aid with fairness research.

### D.2.3 Implementation

**Vendor transparency.** As third-party vendors offer an alternative path for data collection, we recommend curators prioritize transparency both in negotiations with these vendors and when reporting their results. In negotiations with data vendors, curators should prioritize transparency clauses in the contract. For example, curators should advocate for transparency in data worker identities and compensation handling. This can help to ensure that they have access to necessary information for evaluating dataset fairness. During the collection process, data vendors should be held accountable for transparency practices through regular monitoring and evaluation. This could involve conducting audits or assessments to ensure compliance with transparency agreements and guidelines.

To reduce the burden on individual curators, there should be a discipline-wide effort to evaluate and benchmark data vendors based on transparency and ethical data collection practices. From management studies, there is a line of work on vendor evaluation systems and vendor scorecards that can be adapted for third-party data curation services [95].

**Language barriers.** When faced with language barriers, the data curation team should ensure that they have members who have an understanding of the data collection project's context, goals, and data requirements such that they can provide more contextually appropriate translations. If this is not possible, we recommend establishing partnerships with local community organizations or language schools to access language resources at reduced costs.

**Fair data labor.** To ensure fair data labor practices, we recommend curators create clear guidelines and protocols for hiring, training, and evaluating data workers to promote fairness and prevent exploitation. Following prior works [3, 26, 147, 155], we also advocate for transparent and equitable compensation structures for data workers. When possible, curators should provide opportunities for professional development and advancement for data workers.

**Diverse data availability.** Curators should consider using alternative data sources beyond web data, such as community-driven platforms or public repositories, to supplement dataset diversity. To support this, organizations should invest in creating public data trusts [27] or data consortia [80] as an alternative source for large-scale data.

**Data collector availability.** To address a lack of data collector availability, we recommend curators form partnerships with universities, organizations (e.g., NGOs, non-profits), or community groups, operating in underrepresented regions. These partnerships can help the recruitment of data collectors from the target regions, leveraging existing networks and/or local expertise to overcome challenges. For example, Rojas et al. [122] partnered with Gapminder and individual photographers to collect geographically diverse images for the DollarStreet dataset.

**Data annotator diversity and expertise.** When recruiting data annotators, curators should research and understand which personal attributes are legally protected and cannot be asked about during the hiring process. Further, they should be cognizant of cultural nuances. For example, disclosing sexuality can potentially endanger workers. Thus, rather than directly asking sensitive personal attributes, curators can utilize alternative methods for assessing annotator qualifications and suitability for the project. This is especially helpful when annotators may not want to disclose certain attributes.

Finally, curators should offer training and resources to annotators to help them understand the cultural significance of the data they are annotating.

### D.2.4 Evaluation

**Gold standard paradigms.** Dataset curators can adopt evaluation methods that embrace diverse perspectives rather than only using consensus-based methods, which may only showcase the viewpoint of the majority. Works from both machine learning [33, 90] and human-computer interaction [56, 57] have encourage using a multiplicity of annotations, which can showcase disagreement, rather than using majority voting. For example, a curator may capture a diversity of annotations from each annotator, with qualitative explanations as to why the annotator chose each label. Prior works [116, 150] have also provided frameworks for quantifying disagreement across diverse groups of annotators that can be used as an alternative measure to consensus-based approaches.

**Evaluating immeasurable constructs.** When it comes to evaluating immeasurable constructs, curators can supplement quantitative metrics with qualitative approaches. This could include interviews with data workers to better understand their point of view and reveal any potential biases or ethical issues that arose during the collection process. Furthermore, as Miceli et al. [102] advocate for in their work, there should be more reflexivity in the data collection process. Concretely, refereed publications should require more critical reflection on the limitations and trade-offs of the dataset by the curators.

### D.2.5 Maintenance

**Unstable infrastructural ecosystems.** To manage unstable infrastructural ecosystems, we recommend building standardized methods for checking the availability of data sources and creating protocols to replace instances if they have been deprecated. For example, P8 mentioned developing automated scripts that would periodically check whether their dataset instances were still available. Rather than waiting for dataset users to notify curators that certain instances are no longer available, this allows for proactive maintenance.

Going hand-in-hand with this, once curators are aware that certain instances are deprecated, they should have a plan for replacing them in a way that maintains the overall composition of the dataset. This can be challenging, especially for datasets where compositional fairness is prioritized. We recommend that dataset curators create a protocol for identifying alternative data sources that match the distribution and characteristics of the original dataset at the design phase. For example, dataset curators can keep a portion of collected data as "backup" that they can use to replace instances that are deprecated or removed.

**Dataset traceability mechanisms.** One challenge curators faced was tracing how their dataset was used after release. Often they relied on citation metrics as a proxy; however, it was difficult to disambiguate whether the citation meant the authors were using the dataset or referring to concepts in the paper. As an alternative, we recommend curators require users to register or authenticate their identity before accessing datasets, enabling better tracking and accountability. For example, ImageNet [37] requires users to sign in to their platform before downloading data. Another option can be to use permanent digital identifiers, such as DOIs, which is already a standard for some journals such as *Nature*.[4] Similarly, curators can use centralized data repositories (e.g., Hugging Face, Kaggle, Zenodo, Harvard Dataverse, Mendeley data).

## E   Broader Impacts

Our work focuses on understanding the challenges with fair dataset collection by conducting on-the-ground interviews with dataset curators. We provide a taxonomy of challenges that curators face throughout the dataset lifecycle and an exploration into the broader landscape of challenges curators face. For dataset curators, this work provides valuable insight into the nuance and trade-offs related to dataset creation that may not appear in publications. By formalizing this otherwise tacit knowledge, we hope to make the process of fair dataset collection more accessible for curators. More broadly, we intend for our work to have an impact on machine learning as a discipline. We seek

---

[4]https://www.nature.com/ncomms/editorial-policies/reporting-standards

to emphasize the importance of dataset curators' labor, which often is undervalued [124, 125, 131]. Furthermore, we provide an extensive set of recommendations that can be implemented by either individual contributors or from the top-down. By using these recommendations and the challenges we have surfaced, we hope to help facilitate better fair dataset curation practices within the machine learning community.

## F Author Contributions

D.Z. and J.T.A.A. conceived of the idea for the project in this paper. D.Z., M.S., P.C., J.T.A.A., G.P., S.W., and K.P. were involved in discussing the themes of the overarching project. J.T.A.A. and A.X. acquired the financial support for the project. J.T.A.A., S.W., K.P, and A.X. provided oversight and leadership to the research team working on the project.

D.Z., J.T.A.A., P.C., S.W., and K.P. were involved in developing the interview protocol. P.C., S.W., and K.P. recruited participants and conducted the interviews. P.C. transcribed and redacted the interviews.

D.Z., M.S., J.T.A.A., P.C., G.P., S.W., and K.P. were involved in developing the thematic codebook from the interviews. D.Z., M.S., J.T.A.A., and K.P. conducted analysis of the interviews, applying themes from the codebook.

D.Z. and M.S. drafted the manuscript. J.T.A.A., G.P., S.W., and K.P. reviewed and commented on the manuscript. M.S., P.C., and D.Z. created the figures and tables in the manuscript.

