# OpenReview forum: "A Taxonomy of Challenges to Curating Fair Datasets"
_NeurIPS.cc/2024/Datasets_and_Benchmarks_Track — NeurIPS 2024 Track Datasets and Benchmarks Oral_

### Official Review · Reviewer_jzy5 · 2024-07-22
**Review of A Taxonomy of Challenge to Curating Fair Datasets**

**Rating:** 7
**Confidence:** 3
**Correctness:** The methods and claims of the paper a…

**Review:**

The authors face an exciting and relevant problem by extracting insights from a set of interviews with ML practitioners. However, despite the paper's main contribution being a taxonomy, this is hard to follow and explain across different sections (Is Table 4 part of the taxonomy?). Also, this taxonomy overlaps with existing related work in the field (most of them cited in the paper), but no comparison of this overlapping is made. While the overlapping could be a good contribution because it provides more proof of the existence of a specific issue, I do not see why the sample of the interviews could generate relevant insights for our specific use case.

In summary, while this paper's contribution is interesting, some rework is needed to make the explanation more consistent and to compare it with existing works.

**Strengths:**

The paper is well motivated in a relevant existing problem.

**Additional Feedback:**

Compare the extracted conclusions with the existing works in data reporting and documentation, proposing actionable improvement in these reports, if needed.

**Clarity:**

The structure of the paper is difficult to follow. The taxonomy represents the paper's main contribution, and its explanation is divided into different sections. The taxonomy's resume needs to be consulted in the appendix.

**Documentation:**

Methods and documentation are appropriate.

**Ethics:**

No ethical concerns as the empirical study have been approved by author's university.

**Limitations:**

While challenges during data curation have been widely analyzed in the related work, I do not see why the sample of practitioners provides novel and relevant insights into the problem.

However, despite the paper's main contribution being a taxonomy, this is hard to follow and explain across different sections. For instance, is Table 4 part of the taxonomy, or is it just a further analysis beyond it?

A comparison with related works on data reporting and documentation may be needed to understand this paper's contribution more clearly.

**Opportunities For Improvement:**

Compare the extracted conclusions with the existing works in data reporting and documentation, proposing actionable improvement in these reports, if needed.

**Relation To Prior Work:**

Several works have proposed comprehensive guidelines for reporting and documenting datasets. These works, most of them cited in the paper, have overlapping conclusions, such as the ones drawn on the paper. It will be interesting to point out at least the most relevant overlapping conclusions with prior work that comes up from the gathered interviews.

**Summary And Contributions:**

The paper proposes a taxonomy of challenges and trade-offs for curating a fair dataset extracted from a set of interviews with ML curations. The work provides insights into the interviews in different aspects of the dataset lifecycle, such as requirements, design assumptions, implementation, evaluation, and maintenance. Then, provide insights from the interviews into challenges overarching the broader landscape of fairness, identifying 5 levels of challenges: Individual level, Discipline level, Organization Level, Regulatory Level, and Socio-Political level. Finally, it provides a set of considerations to enable fair dataset curation.

---

> ### Author Rebuttal · Authors · 2024-08-14
>
> ### Taxonomy Clarification
> > However, despite the paper's main contribution being a taxonomy, this is hard to follow and explain across different sections. For instance, is Table 4 part of the taxonomy, or is it just a further analysis beyond it?...The structure of the paper is difficult to follow. The taxonomy represents the paper's main contribution, and its explanation is divided into different sections. The taxonomy's resume needs to be consulted in the appendix.
>
> We appreciate the feedback from the reviewer. Our taxonomy is divided into two parts: the first is documenting the challenges across the dataset lifecycle and the second is the broader landscape of fairness. For individual practitioners, our first taxonomy, which is shown in Figure 1 and Table 3, provides insights into what challenges they may face (and need to address) at different steps of the data collection process.
>
> However, we acknowledge that practitioners are not isolated actors; their decisions when collecting datasets are impacted by the broader environment in which they are situated. This is why we introduce our second taxonomy, which is shown in Figure 2 and Table 4. Here, we chart out the larger landscape around dataset collection, highlighting that there are challenges arising not only at the practitioner level but also from the disciplinary, organizational, regulatory, and socio-political challenges that factor into these individual decisions. Moreover, we also use this second taxonomy to point out that the recommendations we provide cannot be situated only at the individual level. This places disproportionate burden on the researcher and furthermore can fail to address the root of some issues.
>
> To make this clearer to the reader, we will highlight the fact that there are two parts to this taxonomy and provide additional explanation as to how these two taxonomies interact with each other.
>
> ### Comparison with prior works
> > Compare the extracted conclusions with the existing works in data reporting and documentation, proposing actionable improvement in these reports, if needed... A comparison with related works on data reporting and documentation may be needed to understand this paper's contribution more clearly... Several works have proposed comprehensive guidelines for reporting and documenting datasets. These works, most of them cited in the paper, have overlapping conclusions, such as the ones drawn on the paper.
>
> We thank the reviewer for this feedback. As written in the introduction, this work is intended to complement these existing works on data curation and insights from these existing works were used to help develop our interview protocol. By taking a qualitative approach and conducting semi-structured interviews, we are able to gather a “thick description” [1] of practitioner’s behaviors, or in other words, we not only understand their behaviors, which can manifest in the datasets or articles they produce, but also the context behind such behaviors.
>
> A key distinction of our work is that we consider fairness in datasets across three dimensions: the composition, the process of collecting the data, and then the release plus subsequent maintenance. While previous works [2, 3, 4] have considered these aspects in isolation, we discuss how challenges can cut across these definitions of fairness or even bring definitions in tension. For example, we include the example of P2 who wants to have a diverse set of annotators for their dataset but is unable to collect certain attributes since they might be illegal in the annotator’s country. Here, wanting to have fairness in composition (i.e., diversity of annotators) comes into conflict with having a fair or responsible data collection process.
>
> [1] Geertz. "Thick Description: Toward an interpretive theory of culture."
>
> [2] Peng et al. “Mitigating dataset harms requires stewardship: Lessons from 1000 papers.” NeurIPS D&B 2021.
>
> [3] Gebru et al. “Datasheets for Datasets.” CACM 2021.
>
> [4] Díaz et al. “Crowdworksheets: Accounting for individual and collective identities underlying crowdsourced dataset annotation.” FAccT 2022.
>
> ### Paper Contribution
> > While the overlapping could be a good contribution because it provides more proof of the existence of a specific issue, I do not see why the sample of the interviews could generate relevant insights for our specific use case. While challenges during data curation have been widely analyzed in the related work, I do not see why the sample of practitioners provides novel and relevant insights into the problem.
>
> As we discuss in the introduction, while previous works [1, 2, 3] have discussed challenges during data curation, these have been theoretical analyses of the problem and provided guidelines to improve data curation. In comparison to previous works, we take an empirical approach to understand how these challenges may or may not manifest on the ground for practitioners. Given the bias for only reporting positive results and the overall lack of documentation in the dataset creation process, our work sheds insight into the nuanced challenges and trade-offs dataset collectors make. Documentation of these considerations --- from the perspective of the dataset creator themselves --- typically are not included in artifacts. Thus, we are helping make this tacit knowledge more institutionalized, which benefits researchers in the future who may want to undertake similar tasks.
>
> [1] Andrews et al. “Ethical considerations for responsible data curation.” NeurIPS D&B 2023.
>
> [2] Hutchinson et al. “Towards accountability for machine learning datasets: Practices from software engineering and infrastructure.”
> FAccT 2021.
>
> [3] Jo and Gebru. “Lessons from archives: Strategies for collecting sociocultural data in machine learning.” FAccT 2020.

---

> > ### Comment · Reviewer_Tw7t · 2024-08-24
> > **Accepting this paper**
> >
> > I thank the authors for their efforts to address my concerns satisfactorily. I have upgraded my review.

---

> > ### Comment · Reviewer_jzy5 · 2024-08-24
> > **Helpful answer's**
> >
> > I want to thank the author's for their answers. Based on the other reviews and the author's response, the contribution in relation other works in the field seems more clear to me (focusing on extractive qualitative insights).
> >
> > Based on this, I upgraded my overall evaluation.

---

### Official Review · Reviewer_Tw7t · 2024-07-24
**A Taxonomy of Challenges to Curating Fair Datasets**

**Rating:** 8
**Confidence:** 4

**Review:**

The quality, clarity, originality, and significance of this work are evaluated below:
## Quality:
The paper is well-structured, with a clear methodology and detailed findings. The use of semi-structured interviews provides rich qualitative data, and the thematic analysis is rigorous. The authors thoroughly summarize challenges across different phases of the dataset lifecycle—requirements, design, implementation, evaluation, and maintenance. Each challenge is well-documented with participant quotes and contextual explanations, enhancing the reliability and depth of the findings.
However, more could be discussed on how potential biases in participant selection (i.e., how was gender bias addressed in the studied participants?) and interview responses were mitigated. Also, the authors should state how conflicts (inter-rater reliability) were negotiated; this could shed light on the taxonomy's trust and reliability.
## Clarity:
The paper is generally well-written, with clear explanations of concepts and findings.  The use of participant quotes effectively illustrates the points made and provides a grounded understanding of the challenges discussed. The taxonomy is presented in a structured manner, making it easy for readers to follow and understand the various challenges and their implications.
The figures and tables effectively illustrate key points. Some sections, particularly the methodology, could benefit from additional detail to enhance transparency and reproducibility.

## Originality:
The study addresses a significant gap in the literature by focusing on practical challenges in dataset curation, which has received less attention than algorithmic fairness. The taxonomy of challenges is a novel contribution that provides a structured framework for understanding and addressing fairness issues in dataset curation.

## Significance:
The findings have significant implications for both researchers and practitioners. By highlighting the practical challenges and proposing actionable recommendations, the paper contributes to ongoing efforts to improve fairness in ML. The study’s insights can inform the development of better practices and policies for dataset curation.

## Pros and Cons:
### Pros:
- A comprehensive taxonomy of challenges.

- Rich qualitative data from diverse participants.

- Actionable recommendations for improving fairness.

- Clear and informative figures (1 & 2) and tables (3 & 4).

### Cons:
- There is limited discussion on mitigating biases in the qualitative study.

- Some sections could benefit from more detailed explanations.

- There are no clear links between the dataset curation lifecycle and the three dimensions of fairness.

- There is a lack of empirical evidence to support some recommendations.

- Unfortunately, this study did not satisfy two of the three dimensions of fairness. I don’t see how composition and process were respected/materialized.

- The generalizability of this work could be a potential threat due to the nature of such qualitative studies. However, the authors should indicate how quantitative analysis could build on the taxonomy constructs for validation.

**Strengths:**

## Strengths
1. **Significance of Contribution**: The taxonomy of challenges is a significant contribution to the field, providing a structured framework that can guide future research and practice.
2. **Relevance to Broader Research Community**: The paper addresses an important and timely issue relevant to stakeholders such as researchers and practitioners working on ML fairness.
3. **Quality of Research**: The use of semi-structured interviews and thematic analysis is appropriate and well-executed, providing deep insights into practical challenges.
4. **Ethical and Social Implications**: The focus on fairness and the ethical considerations in dataset curation are commendable, highlighting the social impact of the work.

**Additional Feedback:**

## Additional Feedback
- **Detailed Methodology**: Consider providing more detail on the thematic analysis process to enhance transparency.
- **Empirical Validation**: Incorporate empirical evidence or case studies to support the recommendations.
- **Bias Mitigation**: Discuss potential biases in participant selection and interview responses and how they were addressed.

### Questions for Authors
1. Can you provide more detail on how participants were selected and how potential selection biases were mitigated?
2. How did you ensure the reliability and validity of the thematic analysis?
3. Can you provide empirical evidence or case studies to support the recommendations made in the paper?

**Clarity:**

## Clarity
The paper is well-written, with clear and concise explanations. The use of figures and tables enhances understanding. Some sections, particularly the methodology, could benefit from additional detail.

**Correctness:**

## Correctness
The claims made in the submission are generally correct. The qualitative data is analyzed soundly, and the thematic analysis is appropriate. However, more empirical validation of the recommendations would be beneficial.

**Documentation:**

## Documentation
No additional documentation was provided besides the 35 pdf pages with a detail explanation of the dataset curation process.

**Ethics:**

## Ethics
The study adheres to ethical standards, with approval from Arizona State University’s Institutional Review Board. There are no significant ethical concerns, but further detail on informed consent and data protection would enhance transparency.

**Limitations:**

## Limitations
The authors have adequately addressed many limitations, but there are areas for improvement:
- **Bias Mitigation**: More detail on how biases in qualitative data collection and analysis were addressed would enhance the study's credibility.
- **Empirical Support**: Providing empirical evidence for the recommendations would strengthen the paper's impact.

**Opportunities For Improvement:**

## Opportunities for Improvement
1. **Mitigating Biases**:
   - **Location**: Methodology section (see also Page 21, Appendix A).
   - **Detail**: The paper could discuss potential biases in participant selection and interview responses and how they were mitigated.
   - **Suggestion**: Provide more detail on the participant recruitment process and any steps taken to ensure diverse and representative sampling.

2. **Methodological Transparency**:
   - **Location**: Data Collection and Analysis section (also see Appendix A).
   - **Detail**: Some sections, particularly the methodology, could benefit from more detailed explanations.
   - **Suggestion**: Include more detail on the thematic analysis process, such as coding procedures and theme development.

3. **Empirical Evidence**:
   - **Location**: Discussion and Recommendations sections.
   - **Detail**: Some recommendations lack empirical support.
   - **Suggestion**: Incorporate case studies or empirical data to support the proposed recommendations.

**Relation To Prior Work:**

## Relation to Prior Work
The paper clearly discusses how this work differs from previous contributions. It situates the study within the broader context of fairness research and identifies the unique contribution of an empirical investigation into dataset curation challenges.

**Summary And Contributions:**

## Summary and Contributions

This manuscript presents a comprehensive taxonomy of challenges in curating fair and unbiased datasets for machine learning (ML). Through 30 semi-structured interviews with ML dataset collectors, the study explores the definition of fairness, processes for collecting fair datasets, challenges faced, and trade-offs encountered during dataset collection. The study spans interviews conducted between November 2023 and March 2024 and is approved by the IRB ethics board of Arizona State University.

### Key Contributions:
1. **Taxonomy of Challenges**: The paper presents a comprehensive taxonomy of challenges faced at different stages of the dataset lifecycle. This taxonomy is structured across phases such as scoping, design, and post-collection, providing a detailed understanding of the multifaceted issues that arise.
2. **Broader Landscape of Fairness**: It categorizes challenges into a broader landscape, encompassing individual, organizational, and regulatory levels. This multi-layered approach underscores the complexity and interconnectedness of ensuring fairness in datasets.
3. Qualitative Insights: The paper provides qualitative insights into the real-world experiences of ML dataset collectors through detailed interviews. These insights, backed by quotations and examples from participants, not only bring theoretical concepts to life but also shed light on the practical implications of the challenges faced in ML dataset collection.
4. **Identification of Trade-offs**: The study identifies key trade-offs encountered in the pursuit of fairness, offering a nuanced perspective on the balance between various competing factors.
5. **Proposed Solutions**: While primarily focused on identifying challenges, the paper also discusses potential strategies and solutions to mitigate these issues, guiding future efforts towards more equitable dataset curation practices.

### Importance:
This paper’s findings are paramount and urgent for researchers, practitioners, and policymakers involved in ML and data ethics. By illuminating the complex landscape of challenges in creating fair datasets, the study paves the way for more informed and practical strategies to enhance fairness and mitigate bias in ML applications, thereby contributing to the urgent need for more equitable dataset curation practices.

---

> ### Author Rebuttal · Authors · 2024-08-14
>
> We thank the reviewer for their recognition of our taxonomy as a significant and novel contribution to the field, providing a structured framework for guiding future research and practice. We are also grateful for the acknowledgment of our paper's relevance to the broader research community, the quality of our research methodology, and the focus on ethical and social implications in dataset curation.
>
> ### Mitigating Biases and Methodological Transparency
> > There is limited discussion on mitigating biases in the qualitative study. The paper could discuss potential biases in participant selection and interview responses and how they were mitigated. Provide more detail on the participant recruitment process and any steps taken to ensure diverse and representative sampling.
>
> We appreciate the reviewer’s feedback regarding methodology transparency. In our current paper, we have included discussion on methodology in Appendix A. We agree that these details are important for ensuring the transparency and rigor of our study. If accepted, for the camera-ready version, we will provide more information on our methodology in the main-body of the text and add extended details to the Appendix.
>
> > Can you provide more detail on how participants were selected and how potential selection biases were mitigated?
>
> Appendix A outlines the participant recruitment process, but we will include further details as follows: The initial set of participants that we reached out to were based on a review of existing machine learning datasets that were described as being more fair or less biased. From this initial set of participants, we also recruited via social media (i.e., call for participants on Twitter) and snowball sampling. To mitigate potential biases, we sampled participants to get a broad representation across dataset modality (e.g., text, image) and across industry / academia. However, as discussed in the limitations section, our participant set is still skewed primarily towards those in academia and located in the Global North. While this geographic bias is not intentional, it is also reflective of the skew towards Western perspectives and the concentration of dataset creation in certain institutions within the machine learning field [1].
>
> This approach was carefully designed to capture a wide range of perspectives relevant to dataset curation across different sectors, regions, and demographic backgrounds. By achieving thematic saturation, we ensured that our sample was sufficiently diverse and representative.
>
> [1] Koch et al. “Reduced, Reused and Recycled: The Life of a Dataset in Machine Learning Research.” NeurIPS Datasets and Benchmarks 2021.
>
> > How did you ensure the reliability and validity of the thematic analysis?
>
> We employed several strategies to ensure the rigor of our qualitative work. Many of the questions we used in our semi-structured interviews are grounded in existing literature on challenges in responsible dataset creation [1, 2, 3]. We also conducted two pilot tests of our interview protocol before starting recruitment, allowing us to iterate and refine our instrument.
>
> To analyze our interviews, we followed best practices from grounded theory. After establishing an initial codebook of themes, the research team (N=4) independently coded one of the interviews. We then reconvened and synchronously discussed how we coded the interviews and analyzed where we differed when applying codes. After this initial coding round, we again independently coded a second interview and repeated the same process of discussing any disagreements amongst the team before creating a finalized codebook. Only after reaching agreement on the definitions and applications of codes did we split up the remaining interviews amongst the team members. The original interview was also recoded using the updated codebook. Since this process was inductive and focused on drawing larger patterns from the interviews, using metrics such as IRR were not appropriate for measuring reliability [4].
>
> To identify themes from the code, we had each member of the research team first generate themes, with supporting quotations, they observed in the interviews. Then, the research team met synchronously over four sessions to discuss and distill these observations into the higher-level themes discussed in the paper.
>
> Finally, to ensure thorough consideration, we drew on a diverse range of expertise by following contemporary interdisciplinary practices [5, 6, 7]. Our team consists of researchers, practitioners, and lawyers with backgrounds in HCI, ML, CV, algorithmic fairness, health sciences and policy, data visualization, and social and behavioral science. With varied ethnic, cultural, and gender backgrounds, we bring together extensive experience in dataset design, model training, and the development of ethical guidelines.
>
> [1] Holstein et al. “Improving fairness in machine learning systems: What do industry practitioners need?” CHI 2019.
>
> [2] Andrews et al. “Ethical considerations for responsible data curation.” NeurIPS D&B 2023.
>
> [3] Gebru et al. “Datasheets for Datasets.” CACM 2021.
>
> [4] Braun & Clarke. Thematic Analysis: A Practical Guide.
>
> [5] Raji et al. “You Can't Sit With Us: Exclusionary Pedagogy in AI Ethics Education.” FAccT 2021.
>
> [6] Romm. “Interdisciplinary practice as reflexivity.” Systemic Practice and Action Research 1998.
>
> [7] Srinivasan et al. “Artsheets for art datasets.” NeurIPS D&B 2021.

---

> > ### Author Rebuttal · Authors · 2024-08-14
> >
> > ### Mitigating Biases and Transparency (cont.)
> > > There are no significant ethical concerns, but further detail on informed consent and data protection would enhance transparency.
> >
> > At the beginning of the interview, participants were asked to provide their informed consent. They were given the option to opt-out of the interview and also told they have the right to withdraw from the study at any time. Participants were also asked for permission to record the study over Zoom. For data protection, each interview was transcribed from the Zoom recording and identifying details — including but not limited to names, institutions, and dataset names — were redacted from the interview transcript before the coding process. The original transcripts and recordings were deleted promptly after the interviews were redacted. Our study protocol was approved by Arizona State University’s Institutional Review Board.
> >
> > While these details are covered in Appendix A, we recognize the importance of making these processes more visible in the main text. We will include a summary of the participant recruitment and bias mitigation strategies in the main text. We will clarify these points in the camera-ready version to ensure the transparency and rigor of our research are fully communicated.
> >
> > ### Dimensions of Fairness
> > > There are no clear links between the dataset curation lifecycle and the three dimensions of fairness.
> >
> > We thank the reviewer for this feedback. The dimensions of fairness come into play across different stages of the curation lifecycle. Considerations around fairness in composition and process typically arise during the requirements, design, and implementation stage. Considerations around fairness in release are more concentrated in the testing and maintenance stage. Nonetheless, this is not a prescriptive delineation. For example, to ensure fairness in the process, practitioners may have to consider what annotator demographic information they want to release in case it contains sensitive information. We will clarify the links between the dimensions of fairness and the curation lifecycle in our text.
> >
> >
> > > Unfortunately, this study did not satisfy two of the three dimensions of fairness. I don’t see how composition and process were respected/materialized.
> >
> > To clarify, not all datasets seek to adhere to all three dimensions of fairness. For example, datasets containing web-scraped images, such as Fairface [1] or Racial Faces in the Wild [2], seek to have fairness in composition by having racially diverse subjects but do not prioritize fairness in process. In fact, as we point out when discussing implementation challenges, factors such as diverse data availability or collector availability can put these definitions in tension.
> >
> > Nonetheless, to illustrate how fairness in composition and process apply to our dataset of interviews, see as follows:
> >
> > 1. **Fairness in Process:** To ensure fairness in the process, as mentioned in Appendix A, we compensated participants $75 USD or the equivalent in their local currency for approximately 45-60 minutes of their time. This compensation rate was in the same range of prior studies [3, 4] that interviewed machine learning practitioners. We also obtained informed consent from participants and made sure it was clear that they had the right to withdraw from the study at any point. Finally, we also redact our transcripts and delete recordings immediately after transcribing to ensure data anonymity; these policies were also communicated to participants orally and in writing via an information sheet.
> > 2. **Fairness in Composition:** When designing our sampling criteria, we focused on having representation across different modalities of high-dimensional, unstructured data (e.g., image and text) and role setting (e.g., academia or industry). We acknowledge that this leads to biases in the geographic distribution of our participants and encourage future research to extend our taxonomy to understanding distinct challenges that may arise from location.
> >
> > [1] Karkainnen and Joo et al. “Fairface: Face attribute dataset for balanced race, gender, and age for bias measurement and mitigation.” WACV 2021.
> >
> > [2] Wang et al. “Racial faces in the wild: Reducing racial bias by information maximization adaptation network.” ICCV 2019.
> >
> > [3] Sambasivan et al. “ “Everyone wants to do the model work, not the data work”: Data Cascades in High-Stakes AI.” CHI 2021.
> >
> > [4] Boyd et al. “Datasheets for datasets help ML engineers notice and understand ethical issues in training data.” CSCW 2021.

---

> > > ### Author Rebuttal · Authors · 2024-08-14
> > >
> > > ### Generalizability of Results
> > > > The generalizability of this work could be a potential threat due to the nature of such qualitative studies. However, the authors should indicate how quantitative analysis could build on the taxonomy constructs for validation.
> > >
> > > We thank the reviewer for this feedback and we will include in discussion of future work how quantitative analysis could complement the qualitative analysis as follows:
> > > 1. **Systematic Literature Review:** Prior works [1, 2] have surveyed the artifacts created by dataset practitioners and provided quantitative analyses. An extension of our work could involve coding papers for mentions of these challenges. However, a significant challenge—and a key limitation of such an approach—is that discussions of challenges and negative results are often absent or underreported in published work. This absence highlights a crucial advantage of our qualitative methodology, which allows us to capture insights and difficulties that are typically overlooked or intentionally excluded from formal publications.
> > >
> > > 2. **Survey of ML Dataset Creators:** Another method to bolster our qualitative research is a large-scale survey of dataset creators. For example, implementing a survey asking participants to identify which of the following challenges they have experienced to quantify their prevalence.
> > >
> > > [1] Scheuerman et al. “Do datasets have politics? Disciplinary values in computer vision dataset development.” CSCW 2021.
> > >
> > > [2] Zhao et al. “Measure Dataset Diversity, Don’t Just Claim It.” ICML 2024.
> > >
> > > ### Empirical Evidence for Recommendations
> > > > Some recommendations lack empirical support + more empirical validation of the recommendations would be beneficial. Incorporate case studies or empirical data to support the proposed recommendations.
> > >
> > > The recommendations provided are grounded in challenges that participants described during the interview process. For example, we recommend providing labels across different taxonomies, such as those used by Groh et al. [1], based on our interviews with P1, P2, P15, and P18 (see Sec. 2.2) who all described having to manage trade-offs when designing singular label taxonomies. Similarly, we recommended more cross-disciplinary engagement based on participants such as P29 who admitted to not knowing about explicit guidelines for fair dataset collection (Sec 3.2). We will provide more explicit grounding between our recommendations and empirical evidence from our interviews.
> > >
> > > In the spirit of similar works published at NeurIPS D&B [2, 3], the scope of this paper is focused on identifying challenges and raising potential recommendations. A fruitful direction for future work is to design interventions for these recommendations and empirically study the effects on fair dataset collection.
> > >
> > > [1] Groh et al. “Towards Transparency in Dermatology Image Datasets with Skin Tone Annotations by Experts, Crowds, and an Algorithm.” CSCW 2022.
> > >
> > > [2] Raji et al. “AI and the Everything in the Whole Wide World Benchmark.” NeurIPS D&B 2021.
> > >
> > > [3] Peng et al. “Mitigating dataset harms requires stewardship: Lessons from 1000 papers.” NeurIPS D&B 2021.

---

### Official Review · Reviewer_ZA9u · 2024-07-25
**Informative Study**

**Rating:** 6
**Confidence:** 3
**Correctness:** Yes.

**Review:**

The paper documents challenges faced in creating fair datasets. The paper summarizes a study that was done by interviewing 30 researchers. The paper provides potential solutions or best practices to address the challenges faced, referencing a series of proposed solutions across disciplines.

**Strengths:**

- The paper is well written and easy to follow. It is very informative.
- The paper breaks the dataset creation process into an intuitive series of layers. The paper discusses challenges across each stage in the dataset lifecycle (creation to maintenance). It also discusses the challenges and potential solutions at each organizational level (individual to regulatory). The recommendations are wide-reaching.
- The paper does a good job of summarizing the study across 30 participants into a series of shared findings across participants.
- The study includes a group of participants across multiple subfields (NLP, CV, etc) industry and academia, and geographic location.
- The paper describes issues that are shared across all groups, and ones that are specific to a specific group (subfield, job type, location). The issues that were specific various groups were informative, as they may reflect issues that unaffected groups may cause without realizing.

**Additional Feedback:**

N/A

**Clarity:**

The paper is very well written and easy to follow. However, it would be nice if the figure documenting the dataset lifecycle and more concrete recommendations were in the main text.

**Documentation:**

N/A

**Ethics:**

No.

**Limitations:**

Yes. The authors discuss the limitations in the sample size and limited number of participants outside of North America or Europe.

**Opportunities For Improvement:**

- The target audience of the paper is limited, many of the proposed recommendations come at the level of regulatory response, and individual dataset curators may find it infeasible to implement many of the recommendations.
- Many of the documented suggestions do not address some of the fundamental issues that study participants surfaced. For example, in section 2.1 “P13 noted ML is “in this age of scale,” making them “a bit skeptical as [to] whether people are going to openly use fair datasets for training unless they’re very large.” It is unclear how many of the recommendations can fit into web-scale datasets - especially ones that are automatically generated.

**Relation To Prior Work:**

Yes.

**Summary And Contributions:**

The paper seeks to educate readers about more ethical and fair dataset collection. The authors interview 30 dataset curators as a study into the creation and maintenance of datasets.

The contributions are:
1) A condensed survey documenting known challenges and tradeoffs among dataset curators.
2) A breakdown into the various issues that affect building a fair dataset during all parts of the dataset development process.
3) Recommendation for best practices to help address the issues that arise at each point in curation.

---

> ### Author Rebuttal · Authors · 2024-08-14
>
> We thank the reviewer for the thoughtful feedback. The recognition of the paper’s contributions, especially the clear breakdown of challenges and recommendations across the dataset lifecycle and organizational levels, is appreciated. It is encouraging that the study’s broad participant base and its relevance to multiple subfields resonated well.
>
>
>
> ### Target Audience Limitations
> > The target audience of the paper is limited, many of the proposed recommendations come at the level of regulatory response, and individual dataset curators may find it infeasible to implement many of the recommendations.
>
> The concern regarding the target audience is acknowledged. Similar to prior works [1, 2], the intention behind the recommendations is to address challenges at both the individual curator level and higher organizational or regulatory levels. This also reflects the many stakeholders involved in dataset creation. We address the reviewer’s comment in two parts:
>
> 1. **Macro-scale recommendations are needed to advocate for systemic change:** Our paper emphasizes the need for systemic changes to facilitate fair dataset curation practices. Expecting individual curators to solve these problems without broader structural support would be unrealistic and could limit the effectiveness of any proposed solutions. Furthermore, recommendations, although targeting a broader audience that includes regulators and organizations, have the potential to create ripple effects that benefit individual curators.
>
> 2. **Awareness around best practices is essential for curators:** We acknowledge the reviewer’s comment that curators may find implementing all recommendations infeasible. In fact, a core thread of our paper is highlighting how practitioners deal with difficult trade-offs during dataset collection. However, as pointed out in Sec. 3.2, one challenge is that practitioners are often unaware of these best practices. Thus, by taxonomizing challenges and providing clear-cut recommendations, we can help ameliorate this issue of awareness, with the understanding that practitioners may not be able to adopt every recommendation in practice.
>
>
> [1] Peng et al. “Mitigating dataset harms requires stewardship: Lessons from 1000 papers.” NeurIPS D&B 2021.
>
> [2] Jo and Gebru. “Lessons from archives: Strategies for collecting sociocultural data in machine learning.” FAccT 2020.
>
>
> ### Challenges with Web-Scale Datasets:
> > Many of the documented suggestions do not address some of the fundamental issues that study participants surfaced. For example, in section 2.1 “P13 noted ML is “in this age of scale,” making them “a bit skeptical as [to] whether people are going to openly use fair datasets for training unless they’re very large.” It is unclear how many of the recommendations can fit into web-scale datasets - especially ones that are automatically generated.
>
> The difficulty of applying fairness principles to large datasets is acknowledged. However, this perspective doesn't fully capture the broader context and intent of our paper. Our recommendations aim to build a foundation for systemic change, which is necessary for making fairness scalable in the future. Our paper acknowledges the difficulty of applying fairness principles at scale today but argues that addressing foundational issues is a critical step toward eventually achieving that goal. More concretely:
>
> 1. **Scale Challenges:** Our paper documents challenges associated with scaling fairness in large datasets through the voices of participants like P13. Our paper's role is to surface these issues, providing a taxonomy of challenges, rather than offering a one-size-fits-all solution. The skepticism about the scalability of fairness is a recognized challenge, and the paper's recommendations are part of an ongoing conversation rather than an absolute solution.
> 2. **Focus on Foundational Issues:** While some recommendations might not seem directly applicable to large-scale datasets, they address foundational issues that must be resolved to achieve fairness at scale. For instance, the emphasis on flexible taxonomies and diverse representation is crucial groundwork that will enable future advancements in scaling fair practices.
> 3. **Recommendations as Part of a Broader Strategy:** The recommendations in our paper are designed to foster systemic changes that can gradually address fairness issues, even in large-scale datasets. While it may be difficult to apply some of these recommendations directly to web-scale datasets today, the intent is to shift the field towards practices that could make such applications more feasible in the future. For example, improved taxonomy design, enhanced traceability, and better tools for dataset curation are steps towards this goal. The recommendations should be seen as part of a broader, adaptive strategy that will evolve as technologies and methodologies develop. Our paper is a starting point, not a final answer.
>
> ### Adding Figures to Main Text
> >It would be nice if the figure documenting the dataset lifecycle and more concrete recommendations were in the main text.
>
> We appreciate the suggestion. We will move the figure to the main text to ensure it is more accessible to readers. Additionally, we will elaborate upon the concrete recommendations currently in the appendix and incorporate this directly into the main text.

---

> ### Author Response · Authors · 2024-08-28
> **Follow-up on discussion period**
>
> Dear Reviewer ZA9u,
>
> As we near the end of the discussion period, we would appreciate any feedback on whether our recent responses have sufficiently addressed your concerns. If there are any further clarifications or additional details we can provide, please let us know!
>
> Thank you for your time and consideration.
>
> Best wishes, Authors of the submission

---

### Official Review · Reviewer_aifA · 2024-07-25
**A comprehensive overview of challenges faced by data curators**

**Rating:** 9
**Confidence:** 4
**Correctness:** The claims are well evidenced.
**Clarity:** Very well written - thank you!

**Review:**

I thought this paper was excellent. Very easy to read and very well structured. I particularly appreciated the different dimensions of data curation through the dataset lifecycle and then separately considering the ecosystem incentives that can make the practical / technical challenges harder to address.

The depth of the interviews was really powerful and the nuances were explained clearly through selected quotes.

I think this paper is of wide relevance and I particularly appreciated the focus - that came from the interviewees - on fair labour practices and pay.

**Strengths:**

This qualiatative research is well conducted and the insights very clearly described. The dimensions of challenges (the taxonomy) are clear and can provide a foundation for practitioners, funders, and policy makers to consider the impact of the incentives they perpetuate.

**Additional Feedback:**

A really great paper - thank you - I enjoyed reading it and learning with you so much!

**Documentation:**

Yes - thank you for sharing the interview protocol in the appendix. I found that very helpful for understanding the details of how the project was completed (and I agree that it is best placed in the appendix!)

**Ethics:**

No ethical concerns.

**Limitations:**

The limitations are clearly stated but they are currently in the supplementary material - please find a way to squeeze them into the main paper.

**Opportunities For Improvement:**

I think the limitations need to be in the main paper. This project has a bias towards western perspectives and that is itself an "unfair" dataset.

I don't think the limitations detract from the value of the paper - but they should be acknowledged more clearly.

Otherwise I think the only suggestion I have is to think about whether there is a call to action for data curators to adapt the taxonomy - how could you increase the dataset - particularly thinking about bringing in additional expert voices from the Global South?

**Relation To Prior Work:**

The background and context is in the appendix and so less likely to be digested by a regular reader. But I think the additionality of this qualitative work is clearly motivated there. I also agree as an expert reviewer that there is a long way to go to understand how guidelines and principles can be operationalised in practice. This paper's focus on spanning the whole data lifecycle AND the many levels of ecosystem incentives and challenges is very comprehensive.

**Summary And Contributions:**

This submission synthesises interviews with 30 dataset curators into a taxonomy of challenges that they encounter. The taxonomy of fairness is multi-faceted and curated into three dimensions: composition, process and release. The authors report on findings from the interviews across the requirements, design, implementation, evaluation and maintenance phases of a ML project. They also assess the challenges at different levels within the broader landscape of fair data science: the individual, discipline, organisation, regulatory and socio-policital levels. Finally the paper presents some considerations for enabling fair dataset curation.

---

> ### Author Rebuttal · Authors · 2024-08-14
>
> We appreciate the reviewer's positive and encouraging feedback. We are pleased that the paper’s structure, insightful interviews, and focus on fair labor practices resonated well. Additionally, we value the reviewer’s recognition of our comprehensive approach in addressing the data lifecycle and ecosystem challenges, and we are encouraged by the belief that our work can serve as a foundation for practitioners, funders, and policymakers to consider the impact of the incentives they perpetuate.
>
>
> ### Incorporating Limitations into the Main Paper
> > I think the limitations need to be in the main paper. This project has a bias towards western perspectives and that is itself an "unfair" dataset. I don't think the limitations detract from the value of the paper - but they should be acknowledged more clearly.
>
> ​​The reviewer's point regarding the importance of incorporating the limitations into the main paper is well-taken, and these will be included in the camera ready version if accepted. Additionally, the potential bias introduced by the Western perspective is acknowledged, and there will be a discussion on how future research can address this by incorporating more diverse voices, particularly from the Global South. While the bias in our research is not intentional, it reflects the broader dominance of the West and large institutions in the field of ML [1].
>
> [1] Koch et al. “Reduced, Reused and Recycled: The Life of a Dataset in Machine Learning Research.” NeurIPS Datasets and Benchmarks 2021.
>
>
> ### Call to Action for Data Curators
> > Otherwise I think the only suggestion I have is to think about whether there is a call to action for data curators to adapt the taxonomy - how could you increase the dataset - particularly thinking about bringing in additional expert voices from the Global South?
>
> The suggestion to include a call to action for data curators is appreciated, and a section will be added to the conclusion or discussion explicitly encouraging data curators to adapt and utilize the taxonomy. For example, future research can involve more voices from the Global South, possibly through directly collaborating with local institutions, to further enrich the perspectives captured in our work.

---

> > ### Comment · Reviewer_aifA · 2024-08-31
> > **I remain excited and very supportive of this work**
> >
> > With my apologies for such a late reply (at the end of the summer vacation!) I acknowledge the response from the authors.
> >
> > Thank you for incorporating more around the limitations in the "camera ready" version. As my scores were high already I don't intend to change them.

---

> ### Author Response · Authors · 2024-08-28
> **Follow-up on discussion period**
>
> Dear Reviewer aifA,
>
> If there are any additional comments or questions you have, we’d be happy to address them before the discussion period ends.
>
> Thank you for your time and consideration!
>
> Best wishes, Authors of the submission

---

### Author Response · Authors · 2024-08-23
**Rebuttal Submissions**

Dear AC and reviewers,

We understand that the reviewers have busy schedules, and we sincerely hope you could take a moment to review our responses. We have carefully addressed the main concerns in detail. Please feel free to let us know if you have any further questions—we are more than willing to address them to the best of our abilities. Thank you very much!

Best wishes,
Authors of the submission

---

### Decision · Program_Chairs · 2024-09-26

**Decision:**

Accept (Oral)

**Comment:**

The paper presents a summary of challenges and tradeoffs experienced by dataset curators. It covers different stages of building fairness benchmarks and different dimensions including often overlooked ecosystem incentive, fair labor practices and pay. The authors provide recommendations for best practices.
The reviewers appreciated the qualitative study, the depth of the conducted interviews, the contributions, and clarity of presenting the findings.
The authors provided a convincing rebuttal addressing some of the secondary critique of the reviewers.
The paper would be of interest to a wide audience within and beyond the Datasets and Benchmarks track.